# Structure of inhibitor-bound mammalian complex I

Hannah R. Bridges [1,6], Justin G. Fedor [1,6], James N. Blaza [1,5,6], Andrea Di Luca [2,3,6], Alexander Jussupow [2], Owen D. Jarman[1], John J. Wright[1,4], Ahmed-Noor A. Agip[1], Ana P. Gamiz-Hernandez [2,3], Maxie M. Roessler [4], Ville R. I. Kaila [2,3 ✉] & Judy Hirst [1 ✉]

Respiratory complex I (NADH:ubiquinone oxidoreductase) captures the free energy from oxidising NADH and reducing ubiquinone to drive protons across the mitochondrial inner membrane and power oxidative phosphorylation. Recent cryo-EM analyses have produced near-complete models of the mammalian complex, but leave the molecular principles of its long-range energy coupling mechanism open to debate. Here, we describe the 3.0-Å resolution cryo-EM structure of complex I from mouse heart mitochondria with a substrate-like inhibitor, piericidin A, bound in the ubiquinone-binding active site. We combine our structural analyses with both functional and computational studies to demonstrate competitive inhibitor binding poses and provide evidence that two inhibitor molecules bind end-to-end in the long substrate binding channel. Our findings reveal information about the mechanisms of inhibition and substrate reduction that are central for understanding the principles of energy transduction in mammalian complex I.

[1] The Medical Research Council Mitochondrial Biology Unit, University of Cambridge, The Keith Peters Building, Cambridge Biomedical Campus, Hills Road, Cambridge CB2 0XY, UK. [2] Center for Integrated Protein Science Munich (CIPSM) at Department of Chemistry, Technische Universität München, 85748 Garching, Germany. [3] Department of Biochemistry and Biophysics, The Arrhenius Laboratories for Natural Sciences, Stockholm University, 106 91 Stockholm, Sweden. [4] Department of Chemistry, Molecular Sciences Research Hub, Imperial College London, White City Campus, London W12 0BZ, UK. [5] Present address: Department of Chemistry, University of York, Heslington, York YO10 5DD, UK. [6] These authors contributed equally: Hannah R. Bridges, Justin G. Fedor, James N. Blaza, Andrea Di Luca. ✉email: ville.kaila@dbb.su.se; jh@mrc-mbu.cam.ac.uk

Mitochondrial complex I (NADH:ubiquinone oxidoreductase)[1] is central to oxidative phosphorylation in mammalian cells. It captures the free energy released from reduction of ubiquinone by NADH to pump protons across the inner mitochondrial membrane, support the proton-motive force and power ATP synthesis. As an essential respiratory enzyme, and an important contributor to cellular oxidative stress, complex I dysfunctions resulting from mutations in its subunits and assembly factors cause a diverse set of inherited neuromuscular and metabolic diseases[2].

Mammalian complex I comprises 45 subunits: fourteen "core" subunits, which are conserved from bacteria to humans and sufficient for catalysis, and 31 "supernumerary" subunits, which are required for assembly, stability, regulation, or fulfil independent metabolic roles[1,3]. Understanding of the structure of mammalian complex I has leapt forward in recent years due to the advent of high-resolution single particle electron cryomicroscopy (cryoEM). The structure of complex I from mouse heart mitochondria was described recently at 3.3 Å resolution[4] and that of the yeast *Yarrowia lipolytica* at 3.2 Å[5]. The structures illustrate how, as shown previously in the enzyme from *Thermus thermophilus*[6], electrons enter from NADH oxidation, at the top of the hydrophilic domain, and are transferred towards the membrane along a chain of iron-sulphur (FeS) clusters. The terminal cluster, a [4Fe-4S] cluster known as N2, then donates the electrons to ubiquinone-10 ($Q_{10}$). The hydrophobic $Q_{10}$ enters the enzyme from the mitochondrial inner membrane through a long binding channel, elevating its redox active headgroup out of the membrane plane to within electron-transfer distance of N2. Although ubiquinone binding, reduction, and dissociation are now beginning to be defined by structural and functional data[5,7,8], the mechanism that couples the redox reaction to proton translocation remains poorly understood.

The quinone binding tunnel in complex I is long and heterogeneous; the top and bottom sections are hydrophobic, while the central section is surrounded by many charged residues[6,8]. The charged region may be important in linking redox catalysis to proton translocation, because it sits at the start of a chain of charged residues that leads into the membrane plane[4,6,9], and involves the loops from membrane-bound subunits ND1 and ND3 that may move during catalysis[10]. The structures of the mammalian enzyme in both its active and deactive states depict how these loops become disordered when complex I converts from its ready-to-catalyse "active" state to its "deactive" state. The deactive state is a pronounced resting state that forms spontaneously at physiological temperatures in the absence of turnover, and requires reactivation by both NADH and ubiquinone in order to re-enter the catalytic cycle[3,4,11].

The large, heterogeneous and conformationally-labile nature of the ubiquinone binding site may explain, why such a wide variety of compounds with little resemblance to the substrate inhibit complex I[12–14]. However, only limited structural information is available on their bound states. Density for piericidin was shown pictorially in the structure of *Thermus thermophilus* complex I[6], but neither the model nor data were made available, precluding evaluation of the information. The inhibitor 2-decyl-4-quinazolinyl amine has been observed with its headgroup part way up the ubiquinone binding channel of complex I from *Y. lipolytica*[10]. For several families of inhibitor, including piericidin[12,15] extensive structure-function studies have been undertaken, but the information is difficult to interpret without detailed knowledge of the inhibitor-binding mode. No structures of inhibitors bound to the mammalian enzyme have been presented, despite their inherent biomedical interest[16,17], and that inhibitor-bound structures present unrivalled opportunities to access different mechanistically-relevant enzyme states.

Piericidin is a natural insecticide that was first isolated in the 1960s from the spore forming bacterium *Streptomyces mobaraensis*[18], and has now been synthesised chemically[19]. It is a tight-binding complex I inhibitor[20] that resembles a short chain ubiquinone ($Q_3$). The headgroup resembles the ubiquinone headgroup (see Fig. 1), except one of the two carbonyls (reduced to hydroxyls in ubiquinol) is replaced by a 4-pyridone nitrogen. The hydrophobic tail contains an initial isoprenoid followed by two isoprenoid-like groups, with a hydroxyl group close to its end. Piericidin has been described to compete for the same or overlapping binding sites as the inhibitors rotenone and DQA[21], and to display partially-competitive inhibition with the substrate analogue $Q_2$[22] but mixed behaviour towards decylubiquinone

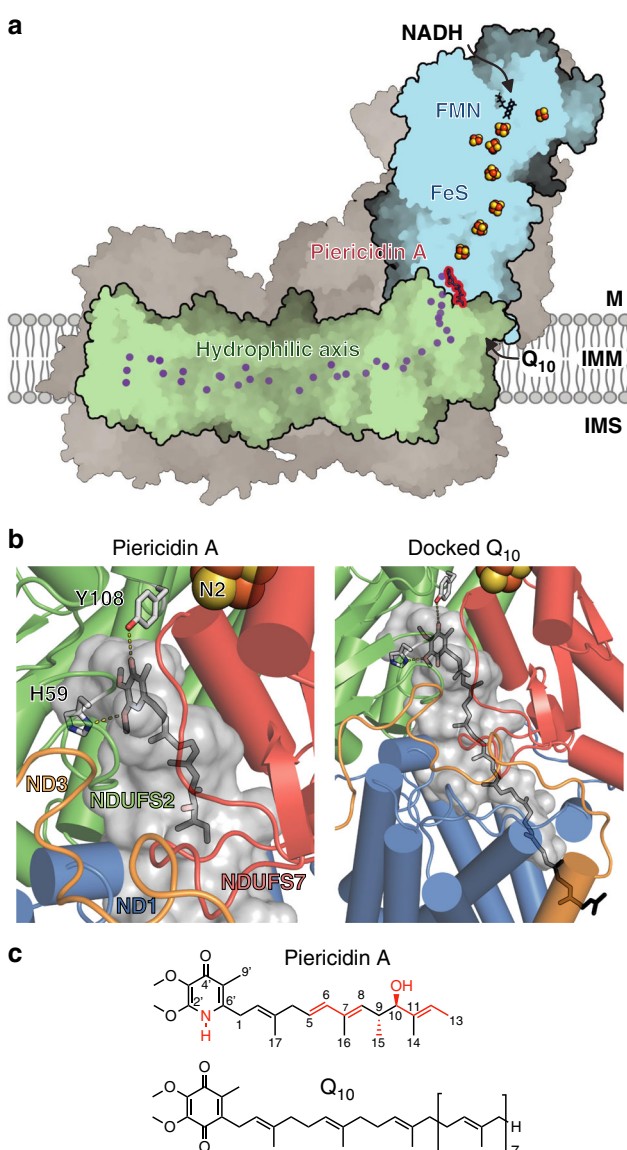

**Fig. 1 Piericidin bound in the structure of mammalian complex I. a** The piericidin molecule is located at the interface between the redox (blue) and proton-transfer (green) domains; supernumerary subunits in grey. Charged residues in the proton-transfer domain are marked in purple. M matrix, IMM inner mitochondrial membrane, IMS intermembrane space. **b** Piericidin (left) and a docked-in $Q_{10}$ (right) in the proposed ubiquinone-binding cavity, leading from Tyr108 and His59 to outside the protein. The internal surface of the cavity was identified using PyMol-1.8.4.0. NDUFS7 helix 4 (residues 104–117) has been removed for clarity. **c** The structures of piericidin and $Q_{10}$ with differences highlighted in red.

(DQ)[23]. It inhibits both the forward NADH:ubiquinone oxidoreductase reaction, and reverse electron transfer (succinate-driven reduction of NAD[+] in submitochondrial particles) with equivalent efficacy[24]. [14]C-labelled piericidin studies describe two non-cooperative binding sites for piericidin for complex I in its native lipid environment, but only one in the delipidated enzyme[24,25].

Here, we have used the canonical complex I inhibitor piericidin A1 (referred to hereon as piericidin for simplicity) to determine the structure of complex I with a ubiquinone-analogue inhibitor bound in its active site. We combine structural, kinetic, spectroscopic, and computational analyses to elucidate the inhibitor-binding mode and the relationships between piericidin and ubiquinone binding, and find evidence for a distal ubiquinone/inhibitor binding site in the central region of the quinone-binding channel. Our piericidin-bound structure is the highest resolution structure of mammalian complex I in an active-like state so far reported, and it reveals information about the mechanisms of inhibition and ubiquinone reduction.

## Results

**Determination of the structure of piericidin-bound complex I.** Piericidin-inhibited complex I was prepared by adding piericidin to the enzyme during turnover with NADH and DQ to ensure exposure of the inhibitor-binding state. Residual substrates and inhibitor were then removed to ensure any inhibitor present was specifically bound. The sample of mouse complex I analysed by

cryoEM was $89 \pm 3\%$ inhibited by comparison with a control sample prepared identically but without inhibitor. The complex was frozen onto PEGylated gold grids and two datasets (Supplementary Table 1) with similar pixel sizes, and numbers of micrographs were collected on FEI Titan Krios microscopes, using either a Gatan Quantum K2 Summit detector with energy filter (piericidin-K2) or an FEI Falcon III detector in counting mode (piericidin-FIII). Both datasets were processed by RELION-3.0 to 3.0 Å resolution (Supplementary Figs. 1, 2, 3 and Supplementary Table 1). Once sharpened, the two maps are essentially indistinguishable: the piericidin-K2 map was used unless otherwise stated. The dataset for the active state of mouse complex I (referred to as active complex I) described previously[4], which reached 3.3 Å resolution with RELION-2.1, was then reprocessed with RELION-3.0 to 3.1 Å (Supplementary Figs. 2, 3 and Supplementary Table 1). The resolutions achieved enabled confident modelling of 97% of the 8430 residues of both the piericidin-bound and active enzymes (Supplementary Table 2).

**Piericidin binding at the top of the ubiquinone-binding channel.** Density for a bound piericidin was readily identified at the top of the ubiquinone-binding cavity, where the ubiquinone ring and first three isoprenoids of ubiquinone-10 are expected to bind (Figs. 1 and 2a, b). Overall, the piericidin-bound maps and model match those for active complex I very closely (98% correlation between the piericidin-bound and active maps

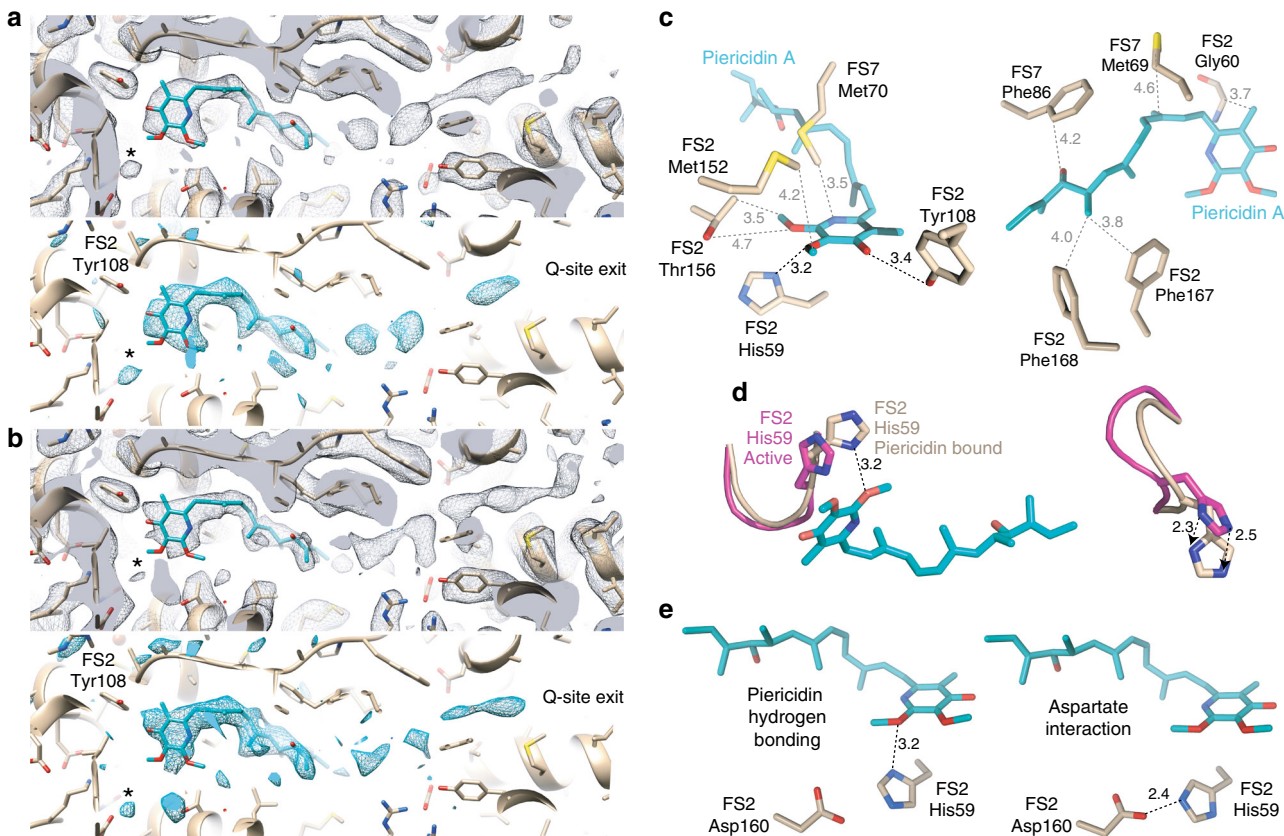

**Fig. 2 Cryo-EM densities for the piericidin molecule, and interactions between the protein and the bound piericidin. a, b** Cryo-EM densities for the piericidin-containing region (black mesh, top) and the difference between the piericidin-free protein model and the piericidin-bound density (teal mesh, calculating using the phenix.real_space_diff_map command) for **a** the piericidin-K2 and **b** the piericidin-FIII datasets. A putative water molecule can be observed in the lower left of each panel (marked by an asterisk). Densities rendered in UCSF Chimera at thresholds of **a** 0.0163, 6.07 and **b** 0.0294 and 5.82. **c** Residues around the piericidin; proposed hydrogen bonds marked in black, other distances in grey. **d** Movement of the NDUFS2 β1-β2 loop between the active (magenta) and piericidin-bound (wheat) models. **e** Interactions of NDUFS2-His59 with the piericidin headgroup and Asp160, depending on the rotation of the His ring; refined structure with interaction with piericidin 2′ methoxy, and ring flipped to form interaction with Asp160. All distances in Å. Subunits NDUFS2 and NDUFS7 are abbreviated as FS2 and FS7, respectively.

in UCSF Chimera and all atom RMSD = 0.6 Å, relative to 2.7 Å for the deactive state (PDB:6G72)) and exhibit all the hallmarks of it: the extended interface between NDUFA5 and NDUFA10 that defines the arrangement of the hydrophilic and membrane domains[4]; clear densities for the mobile loops in NDUFS2, ND3, and ND1[3,4,11]; and α-helical structure throughout ND6-TMH3[4]. Two short stretches of NDUFS7 (residues 58–60 and 86–92, where R91 is known to be hydroxylated)[26] also match the active state. Therefore, we observe piericidin bound to the active state of complex I, in which the ubiquinone-binding site is fully configured and ready for catalysis.

The piericidin headgroup binds adjacent to NDUFS2 His59 and Tyr108, the two proposed ligands of the ubiquinone headgroup (Fig. 2c). Its ring carbonyl accepts a hydrogen bond from the Tyr108 hydroxyl (O–O distance 3.4 Å), consistent with the keto form of the piericidin headgroup (rather than the tautomeric enol form with an aromatic ring, a deprotonated pyridone-N and a hydroxyl). Because piericidin lacks a second carbonyl it cannot form a second ubiquinone-like hydrogen bond to His59. Instead, His59 Nδ1 forms a hydrogen bond (N–O distance 3.2 Å) with the piericidin 2′ methoxy (Fig. 2c). Alternatively, a good fit to the density can be obtained from flipping the imidazole ring to form a hydrogen bond (albeit with poor geometry) between His59 Nε2 and Asp160 (N–O distance 2.4 Å) (Fig. 2e). To accommodate the piericidin, His59 has shifted by ~2.5 Å (Fig. 2d); this is the only difference that could be identified between the piericidin-bound and active structures.

Additional residues around the piericidin may further stabilise it. In particular, NDUFS7-Met70 and NDUFS2-Met152 point towards one face of the ring (Fig. 2c) and the sidechain of NDUFS2-Thr156 is modelled 3.5 Å from the 2′ piericidin methoxy, and could be rotated to bring its hydroxyl group into a hydrogen-bonding configuration (but compromising the fit to the density). Interestingly, weak densities observed between the Thr156 hydroxy, the piericidin methoxy groups, and NDUFS2 Lys371 may arise from bound water molecules and reflect a wider hydrogen-bonded network for ubiquinone protonation/reduction (see below). However, we are not sufficiently confident to model water networks at the current resolution. The piericidin isoprenoid-like tail tracks along the proposed ubiquinone-binding channel, overlapping the predicted positions of isoprenoids one to three (Figs. 1, 2) and surrounded by a series of hydrophobic sidechains (Fig. 2c), including NDUFS7 Phe86 in a π–π interaction and NDUFS2-Phe167 and Phe168 framing the final isoprenoid-like unit. The sidechains of NDUFS7-Thr59 and ND1-Glu204 are near to the hydroxy group on the piericidin chain, but too far to form hydrogen bonds. Glu204 is on the TMH5-6 loop of ND1 at the start of the "E-channel" that connects the ubiquinone-binding site to charged residues in the membrane domain[4,6].

## Molecular dynamics (MD) of piericidin in the substrate binding pocket.

To probe the piericidin-binding mode further, we performed classical MD simulations, starting from the piericidin-bound cryoEM model. The piericidin was modelled into the density, or docked in using MD flexible fitting (MDFF)[27] with the density as a biasing potential. During the MD simulations (Fig. 3 and Supplementary Fig. 4), the NDUFS2 Tyr108-OH forms a stable hydrogen bond with the piericidin 4′ carbonyl, consistent with the cryoEM model, but at a slightly shorter distance of ~2.9 Å (Fig. 3c). NDUFS2 His59, modelled as the doubly-protonated imidazolium (HisH$^+$) samples a hydrogen-bonded conformation with the piericidin 2′ methoxy, but prefers an alternative conformation, resembling that obtained by flipping the His59 ring as in Fig. 2e (right) and stabilised by van der Waals interactions. The histidine further forms an ion-pair interaction with Asp160 (Fig. 3c). Interestingly, simulations with a neutral histidine (Nδ-protonated only) resulted in His59 and the

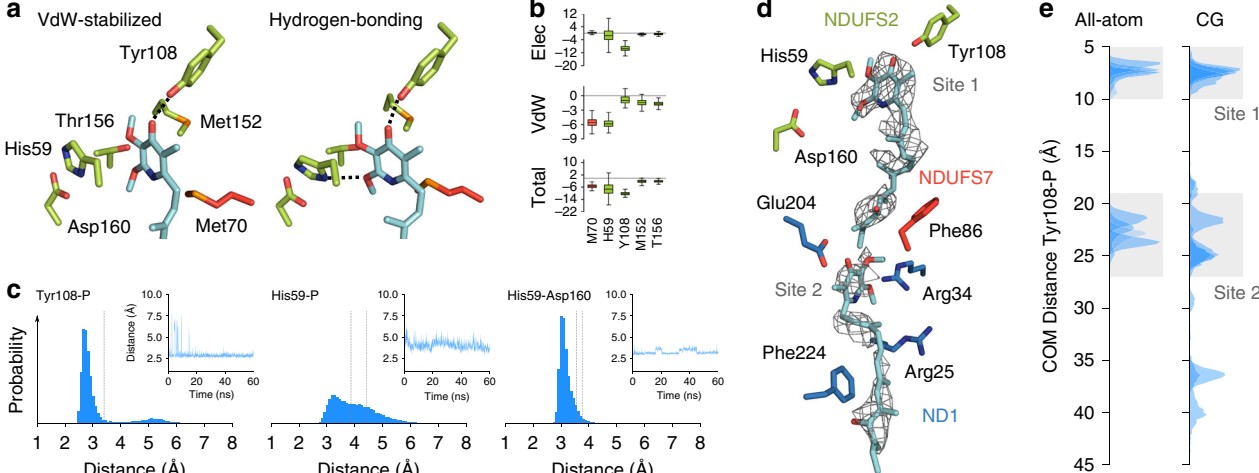

**Fig. 3 Molecular dynamics simulations of piericidin in complex I. a** Snapshots from the MD simulations of piericidin in van der Waals (VdW)-stabilised and hydrogen-bonded conformations. Hydrogen bonds are indicated by dotted lines. **b** Interaction energies between piericidin and selected surrounding residues (total, van der Waals, and electrostatic contributions in kcal mol$^{-1}$). Green, residues in NDUFS2; red, residues in NDUFS7. The boxes extend from the 25th to the 75th percentile, the middle line represents the median. The whiskers show the range of the data from the 10th to 90th percentile ($n = 1500$, snapshots calculated every 40 ps). **c** Distance distributions (simulations A1–A4, Supplementary Table 4) and time traces of distances (simulation A1) between piericidin (P) and key residues. Left: piericidin 4′ carbonyl oxygen to Tyr108 hydroxyl oxygen. Middle: piericidin 2′ methoxy oxygen to His59 (centre of mass of Nδ/Cε/Nε). Right: His59 (centre-of-mass of Nδ/Cε/Nε) to Asp160 (centre-of-mass of carboxylic group). Experimental distances (Fig. 2c) for the refined structure (dark grey) and ring-flipped model (light grey) are shown as dotted lines. See Supplementary Fig. 4 for further time traces from individual simulations, and distances to Thr156. **d** The two modelled piericidin molecules with the experimental cryo-EM density (PyMol-2.2.3 density level 4.0, carve radius = 1.8) and surrounding residues. **e** Distance distributions of piericidin in the substrate cavity obtained from classical (left, simulation A1–A5, 220 ns total) and coarse-grained MD simulations (right, simulation C1–C10, 100 μs total). Distances are between the centre of masses of the Tyr108 and piericidin rings.

NDUFS2 β1–β2 loop moving away from the piericidin and Asp160, suggesting histidine protonation is required for stable piericidin binding. Hybrid quantum/classical (QM/MM) simulations, in which the binding site was modelled using density functional theory (DFT) polarised by a classical model for the surrounding protein, also revealed the hydrogen-bonded and van der Waals binding poses. In addition, at the expense of atomistic detail but with enhanced statistical sampling, coarse-grained (CG) MD simulations support the piericidin remaining stable in the binding pocket for several tens of microseconds. Finally, energy decomposition analyses based on the MD simulations (Fig. 3b and Supplementary Fig. 5) suggest that His59, Tyr108, and Asp160, as well as NDUFS2 Val424 and Phe425, which move closer to the piericidin ring in some of the simulations, make strong contributions to the interaction energy. Although NDUFS2 Thr156 samples transient hydrogen-bonded interactions with the 2′ methoxy (Fig. 3a and Supplementary Fig. 4d) it contributes only moderately to the interaction energy, along with NDUFS7 Met70 and NDUFS2 Met152 (Supplementary Fig. 5). Commensurate with the additional densities observed in the cryoEM analyses, water clustering analyses (see Supplementary Fig. 5b and see "Methods" section) suggest that this region could form a stable water binding site, stabilised by the nearby conserved residues NDUFS2 Asp107 and Asp422.

**The environment of cluster N2.** During the inhibitor-binding incubation, complex I was exposed to NADH to induce turnover and ensure exposure of the inhibitor-binding site. Inhibitor binding prevents reoxidation by DQ, so to determine if the enzyme was prepared in the reduced state, the same protocol was used to prepare a larger amount of the piericidin-bound bovine complex (inhibited by 72 ± 6%) for EPR analyses. With no extra NADH added to reduce the sample, the signal from reduced FeS cluster N2 was clearly observed in the inhibited sample (Fig. 4a). It was also observed in the turnover control (that was prepared identically but not exposed to piericidin), but not in the matching "nonturnover" control to which NADH had not been added (Fig. 4b). No signals from any other reduced clusters were observed. The reduction of only N2 is consistent with the relatively high reduction potential of this cluster in the mammalian complex[28,29], disfavouring electrons transferring from N2 back to the low-potential flavin where they may slowly escape to $O_2$[30]. The samples were subsequently thawed and NADH added to reduce N2 completely, showing that N2 was 78% reduced in the inhibited sample, 44% reduced in the turnover control, and fully oxidised in the nonturnover control. Therefore, we infer that cluster N2 was predominantly reduced in the piericidin-bound sample used for cryoEM, although the small scale of the cryoEM preparation precludes direct measurement. Strikingly, Fig. 4c shows that the environments of N2 in the piericidin-bound (N2-reduced) and active (N2-oxidised) states of mouse complex I are indistinguishable. Even NDUFS2-His190, which becomes protonated at pH < 7.4 when N2 is reduced[29,31] and has a well-defined density in both maps, does not move. Furthermore, the lack of any substantial structural change "downstream" of N2 in the proton pumping modules is consistent with our previous conclusion[29] that N2 redox cycling is not the coupling reaction that initiates proton translocation.

**Kinetic evidence for the binding of two inhibitor molecules.** Inhibition of complex I by piericidin was studied in proteoliposomes reconstituted with complex I, $Q_{10}$, and an excess of the cyanide-insensitive alternative oxidase (AOX) to recycle the ubiquinol back to ubiquinone[32,33]. Proteoliposomes were prepared with varying ubiquinone concentrations to span the $K_M$

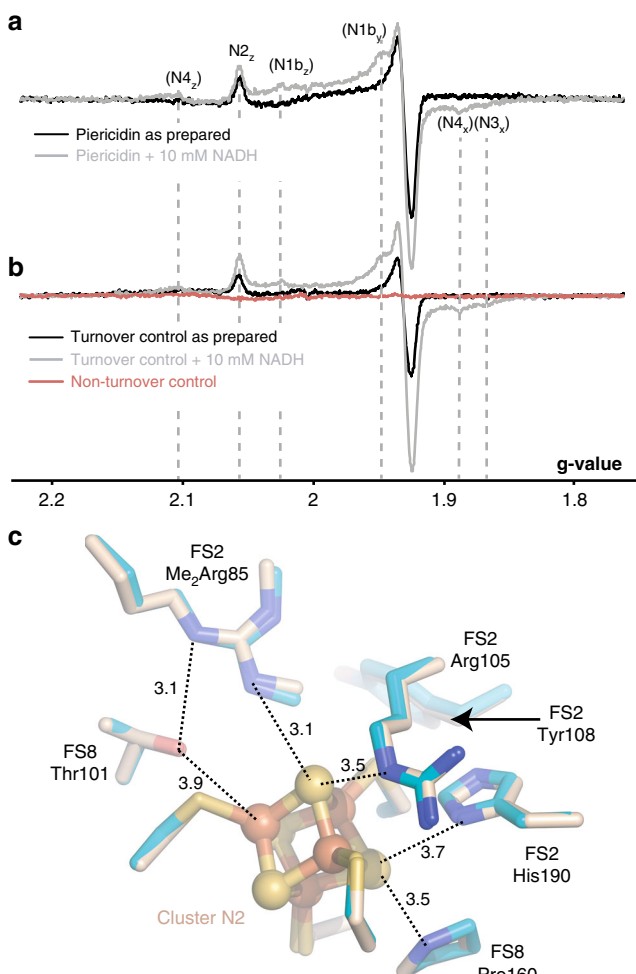

**Fig. 4 Cluster N2 is reduced in the piericidin-bound enzyme but its surrounding structure does not change. a** EPR spectra showing reduced clusters in the piericidin-treated sample as prepared (black) and following addition of 10 mM NADH (grey). **b** EPR spectra showing reduced clusters in the piericidin-free control samples: the turnover control as prepared (black) and following addition of 10 mM NADH (grey) and the nonturnover control as prepared (red). EPR conditions: 9.5 GHz (X-band), temperature 20 K, microwave power 2.0 mW, modulation frequency 100 kHz, modulation amplitude 0.7 mT. **c** Key residues around N2 in the piericidin-bound (teal) and active (wheat) states. Distances are in Å. NDUFS2-Tyr108, the ligand to the piericidin headgroup, is shown for reference.

curve, and characterised by their phospholipid content (hydrophobic phase volume), ubiquinone concentration, and complex I concentration and orientation. Complex I inhibition was then investigated by titrating the rate of NADH oxidation with piericidin. Figure 5 shows that the data can be modelled by a series of Michaelis–Menten curves for their $Q_{10}$ dependence, and a series of $IC_{50}$ curves for their piericidin dependence. Interestingly, both the apparent $K_M$ and $V_{max}$ values decrease with increasing piericidin concentration, contrary to the behaviour expected for a competitive inhibitor (increasing $K_M$ and constant $V_{max}$)— although the piericidin is clearly observed in the binding site for the ubiquinone headgroup in the cryoEM map. Different models were therefore tested as follows. The complex I, ubiquinone-10, and piericidin concentrations were defined relative to the membrane volume, and the ratio of membrane to aqueous piericidin were set by its partition constant ($\log_{10}P = 4.73$). Reoxidation of ubiquinol-10 was assumed to be fast so inhibition by ubiquinol-

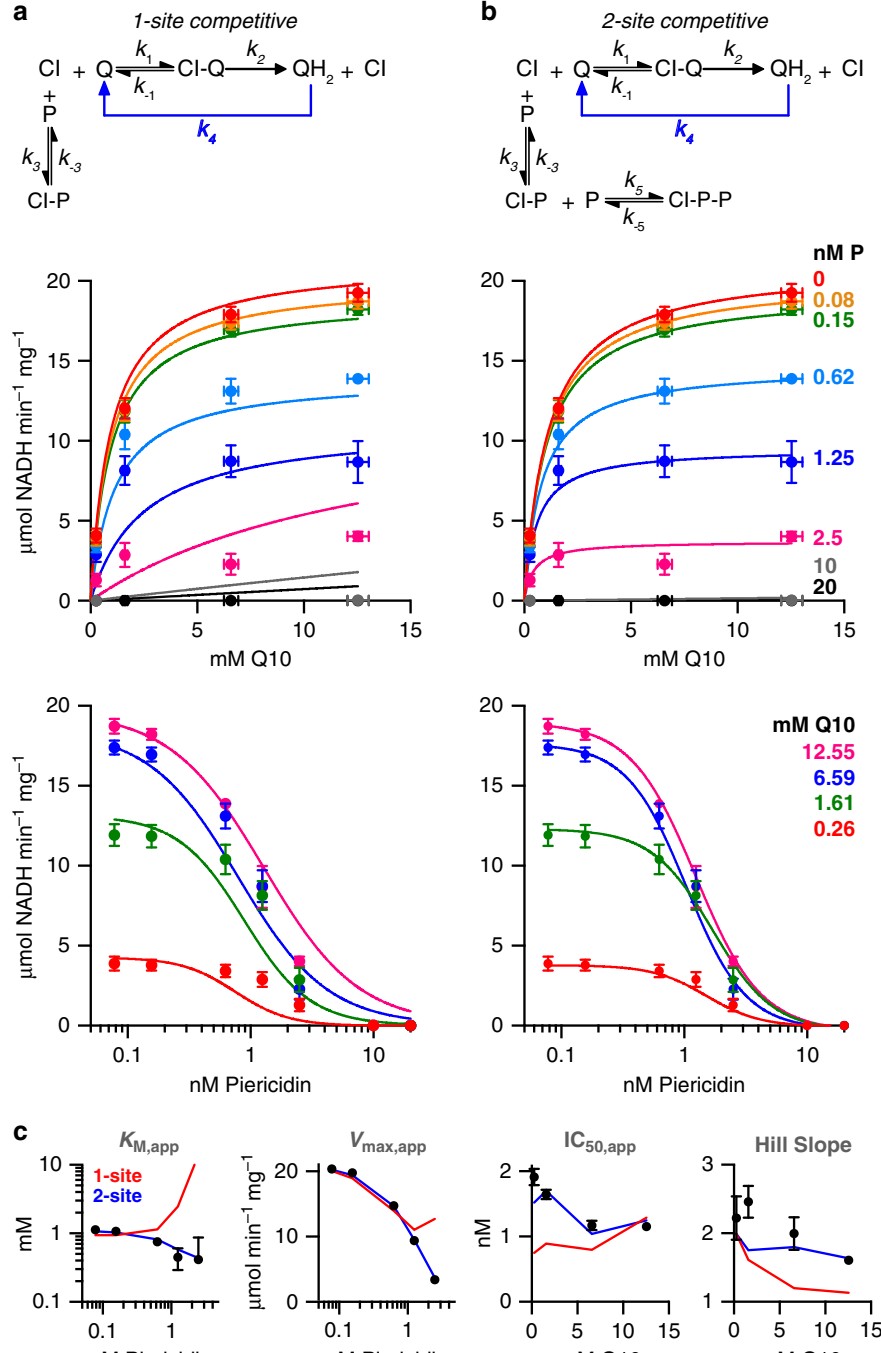

**Fig. 5 Competitive models for piericidin inhibition of complex I in proteoliposomes. a** One-site competitive inhibition mechanism. **b** Two-site competitive inhibition mechanism. In both cases, rapid reoxidation of ubiquinol by AOX ($k_4$) prevents appreciable levels accumulating. Experimentally-measured rates (average ± standard deviation, $n = 3$) are shown in $K_M$ and $IC_{50}$ plots, alongside the best-fit predictions from the models (see Supplementary Table 3 for parameters). **c** Plots of trends in $K_{M,app}$, $V_{max,app}$, $IC_{50}$ and Hill slope produced by fitting $K_M$ and $IC_{50}$ curves to the experimental data (black points, error bars are 95% confidence intervals) and to the output data from the models. The SSR values are 201 and 46.7 for **a** and **b**, respectively.

10 was not considered. Importantly, piericidin is a tight-binding inhibitor, and the pool of ubiquinone/ubiquinol is small, so depletion of their free concentrations upon binding were considered. Although this precluded simple analytical solution of the rate equations and required kinetic rates to be calculated numerically, the numerical approach also enabled us to test essentially any reasonable model without having to make simplifying assumptions, such as about the identity of the rate determining step in complex I catalysis.

Figure 5 confirms that the simplest competitive inhibition model does not account for the data. The sum of the squared-residuals (SSR) for the overall fit (taking into account every data point) is high (Supplementary Table 3) and the observed trends in the apparent values for $K_M$, $V_{max}$, $IC_{50}$, and Hill slope are not replicated. However, by introducing a second binding site to the model, the fit improves substantially: the SSR decreases 7-fold and trends in $K_M$, $V_{max}$, and $IC_{50}$ are predicted well. Early work on the inhibition of complex I by piericidin indicated a

stoichiometry of 2[24], and the Hill slopes of the IC$_{50}$ curves determined here are around 2, suggestive of cooperative binding. Intriguingly, and consistent with the cooperativity, the best-fit $K_{\mathrm{I}}$ values (where $K_{\mathrm{I}}$ is the dissociation constant for the inhibitor) suggest the second inhibitor binds substantially more tightly than the first, with a broad minimum in SSR at $K_{\mathrm{I2}}/K_{\mathrm{I1}} = 10^{-5}$ to $10^{-1}$, and the minimum point at $10^{-4}$, with a 10,000× tighter binding of the second piericidin than the first (see below). Further analysis (Supplementary Table 3) showed that both a single inhibitor binding competitively to the ubiquinol-binding state, and a simple mixed (both competitive and uncompetitive) binding scheme also resulted in poor fits. A two-site uncompetitive binding model, where piericidin binds to the enzyme-substrate complex, was also able to explain the data satisfactorily, but this kinetic model is not supported by the structural data. Therefore, the kinetic data strongly support the two-site competitive inhibition model, implying that more than one piericidin enters the ubiquinone-binding channel.

**Evaluation of densities in the distal section of the ubiquinone-binding channel.** Broken, noisy densities are observed in the "distal" section of the ubiquinone-binding channel in both the piericidin-K2 and piericidin-FIII maps (Fig. 2a, b). However, they are insufficiently resolved for identification—and similar densities are also observed in the active mouse map[4]. To probe whether an additional piericidin would fit the extra densities in the piericidin-K2 map, we performed MD simulations with a second piericidin added. The fitted distal molecule adopts positions that adequately explain the density features observed (albeit with insufficient confidence for inclusion in the cryoEM model), and a hydrogen-bonding interaction was observed to form intermittently between the C10-hydroxyl on the proximal molecule and the 4′ carbonyl on the distal molecule (Fig. 3d). As expected for its location at the position of isoprenoid 4, the headgroup of the distal molecule is also surrounded by a number of charged residues[8]. The charged region, followed by a more hydrophobic region toward the channel entrance, may provide a favourable environment for a range of amphipathic species, and it also corresponds (Fig. 3e) to a recently proposed ubiquinone-binding subsite (site 2) identified in simulations on bacterial complex I[34]. However, the distal piericidin does not adopt a single, well-defined binding pose, and its fluctuations in the MD simulations, together with the unclear density, are better explained by a broad substrate recognition region, rather than a unique, well-defined distal site. The broad recognition region may lead to an entropic stabilisation for the set of subsites to provide a higher affinity for the site as a whole, whereas the interaction between the two piericidins may contribute to the cooperativity, explaining the tighter apparent binding of the second piericidin relative to the first.

Notably, early work to determine the number of piericidins bound to complex I showed there are two nonidentical specific binding sites per membrane-bound enzyme, but that this decreases to one in the delipidated enzyme[24,25]. Loss of high affinity piericidin binding in the detergent-solubilised enzyme, relative to in the membrane, has also been described[22]. Our data are consistent with two piericidins binding in our membrane-bound enzyme assay, and with lower occupancy of the distal site in the detergent-solubilised, structurally-characterised enzyme. Specific phospholipids are known to be essential for complex I activity[35] and their loss and/or the presence of detergent may destabilise the second piericidin, and affect the properties of the distal section of the ubiquinone-binding channel. Although our results thus present an example of how the structurally-characterised detergent-solubilized protein differs from the membrane-bound enzyme, our interpretation is consistent with

an important role for phospholipids, and cardiolipin in particular, in complex I catalysis[35,36].

**Discussion**
Our structures show that piericidin binds at the top of the ubiquinone-binding site between NDUFS2-Tyr108 and NDUFS2-His59, two highly conserved residues long known to be critical for catalysis[37,38]. Tyr108 hydrogen bonds with the piericidin 4′ carbonyl, and our MD simulations suggest it makes a substantial contribution to the binding energy (Fig. 3b and Supplementary Fig. 5). Intriguingly, in the complex I genes present in *S. mobaraensis*, from which piericidin was isolated, a tryptophan replaces Tyr108[39]. There are currently no structures available for ubiquinone bound to complex I, but simulations on the *T. thermophilus* enzyme[40] support an analogous hydrogen bond between Tyr108 and the ubiquinone 4′ carbonyl, poised to protonate the nascent quinol. In the yeast *Y. lipolytica*, all variants of the Tyr (Y144) rendered the enzyme essentially unable to reduce Q$_9$, while Q$_1$ and Q$_2$ supported near wild-type $V_{\mathrm{max}}$ values but with substantially increased $K_{\mathrm{M}}$ values[38]. IC$_{50}$ values for the inhibitor 2-decyl-4-quinazolinyl amine (DQA), measured with Q$_1$, were similarly increased in Tyr mutants[37]—despite structural data showing DQA bound further along the binding channel[10].

Data on the role of His59 in ubiquinone binding and reduction is less conclusive. In piericidin, the 1′ carbonyl is replaced by a pyridone nitrogen hydrogen bond donor (N–H), so His59 must interact differently with piericidin than ubiquinone. Here, the His is observed to hydrogen bond to the 2′ piericidin methoxy. However, our MD simulations suggest piericidin samples two different binding poses, one stabilised by hydrogen bonding and one by van der Waals interactions. Confusingly, while the Ala, Met, or Arg variants of His59 (His95) in *Y. lipolytica* abolished catalysis with no deleterious effects on assembly[38], the analogous NuoD-His224 to Arg variant in *E. coli* gave an enzyme with near wild-type activity with all quinones tested, and near wild-type inhibitor-binding characteristics[41]. The exact interaction mode(s) of His59 with the ubiquinone headgroup and its role in catalysis thus remain unconfirmed.

Variants of NDUFS2-Thr156, NDUFS7-Met70, and NDUFS2-Met152, identified here as relevant to binding, have also been studied in *Y. lipolytica*. Rotenone and DQA inhibit more weakly in *Y. lipolytica* than in the mammalian enzyme[42]. Mutating *Y. lipolytica* Ser192, homologous to mouse Thr156, to Thr increased the affinities for both rotenone and DQA, whereas mutations to Ile, Arg, and Tyr were all detrimental to activity[7]. Variants of Met70 (*Yl* Met91) showed increased $K_{\mathrm{M}}$ values with Q$_1$ and Q$_2$, with $V_{\mathrm{max}}$ hardly affected, and resistance to DQA and rotenone[43]. Similarly, variants of Met152 (*Yl* Met188) exhibited varying amounts of activity with all quinones[42]. Finally, in *Rhodobacter capsulatus* residues on the C-terminal helix of NDUFS2, particularly Val424 (*Rc* Val407) were found to affect piericidin binding[44]. Val424 is close to the 3′ methoxy group on the piericidin headgroup, and is also identified in our energy decomposition analysis (Supplementary Fig. 5).

Our data demonstrate that piericidin competes with ubiquinone for its binding site, and that piericidin binds to an active-like state of mammalian complex I, with all elements of the ubiquinone-binding site defined in the density. However, strictly speaking, the structurally-characterised active state is an off-pathway state with oxidised FeS clusters, because during catalysis NADH oxidation outpaces ubiquinone reduction and cluster N2 is reduced. In contrast, our piericidin-bound structure contains a reduced cluster N2, which does not lead to observable structural changes. Charge delocalisation over the cluster core to minimise reorganisation and facilitate rapid electron transfer, is a feature of

3Fe-4S and 4Fe-4S cluster chemistry. For example, no substantial changes upon reduction were detected in high resolution structures of the 7Fe ferredoxin I from *Azotobacter vinelandii*[45]. In contrast, the NADH-bound, reduced structure of the hydrophilic domain of complex I from *T. thermophilus* was documented to show subtle movements in several helices at the hydrophilic/ membrane domain interface[46]. Corresponding movements are not observed here suggesting they were not representative of the intact enzyme. Furthermore, our density shows no disconnection of either of the tandem cysteine residues that coordinate cluster N2 (Fig. 4b), as described for reduced N2 in the *T. thermophilus* hydrophilic arm, in which N2 is more highly solvent exposed[46].

Our data indicate that two piericidins can be accommodated in the ubiquinone binding channel in the membrane bound complex, with the distal molecule occupying a site that broadly resembles one of the additional binding sites for ubiquinone predicted by simulations on the structure of *T. thermophilus* complex I[34]. First, these sites may represent "staging posts" for the transit of quinone/quinol along the long channel, where the substrate pauses due to favourable interactions with its environment. This staging post concept may help to explain the relatively low $V_{max}$ values of $Q_1$ and $Q_2$[8], since their small size renders them susceptible to becoming trapped at these local energy minima, and as multiple molecules may impede the progress of each other. High concentrations of $Q_2$, in particular, have been reported to inhibit catalysis[12]. The same principles[8] may help to explain why piericidin (which has an alkyl chain length similar to $Q_3$) is so tight binding. The distal site sits within a network of charged residues around the central region of the ubiquinone-binding channel, connected to the E-channel, a series of glutamate residues leading down into the membrane plane. It may therefore be functionally relevant in the (as-yet unknown) mechanism by which ubiquinone binding and reduction is coupled to proton translocation.

## Methods

**Preparation of mouse complex I inhibited by piericidin for cryoEM**. Piericidin-inhibited mouse complex I was prepared by adapting the method of Agip and coworkers for mouse complex I[4]. Six-week old C57BL/6 female mice were sacrificed by cervical dislocation in accordance with the UK Animals (Scientific Procedures) Act, 1986 (PPL: 70/7538, approved by the local ethics committees of the MRC Laboratory of Molecular Biology and the University of Cambridge and by the UK Home Office) and the University of Cambridge Animal Welfare Policy. Hearts were excised, immersed in ice-cold solution containing 10 mM Tris-HCl (pH 7.4 at 4 °C), 75 mM sucrose, 225 mM sorbitol, 1 mM EGTA and 0.1% (w/v) fatty acid-free bovine serum albumin (Sigma-Aldrich), then cut into 1-mm pieces and washed. The tissue was homogenised (10 mL buffer per gram tissue) by seven to ten strokes of a Potter–Elvehjem homogeniser fitted with a Teflon pestle at ~1000 r. p.m then the homogenate was centrifuged (1000 × g, 5 min), and the supernatant recentrifuged (9000 × g, 10 min) to collect the crude mitochondria. The mitochondria were washed several times by resuspension and centrifugation and then collected by centrifugation at 15,000 × g for 2 min, resuspended in 20 mM Tris-HCl (pH 7.4 at 4 °C), 1 mM EDTA and 10% (v/v) glycerol to 10–20 mg protein mL⁻¹ and frozen for storage. After thawing they were diluted to 5 mg protein mL⁻¹, then ruptured by three 5 s bursts of sonication (with 30 s intervals on ice) using a Q700 Sonicator (Qsonica) at 65% amplitude setting and the membranes were collected by centrifugation (75,000 × g, 1 h), resuspended to 5 mg protein mL⁻¹ and frozen for storage.

Starting from 16 mg membrane protein, the membranes were solubilized by addition of 1% *n*-dodecyl β-D-maltoside (DDM, Glycon), stirred for 30 min on ice, and centrifuged (48,000×g, 30 min). The supernatant was loaded onto a 1 mL Hi-Trap Q HP anion exchange column (GE Healthcare) pre-equilibrated with buffer A (20 mM Tris-HCl (pH 7.14 at 20 °C), 1 mM EDTA, 0.1% DDM, 10% (v/v) ethylene glycol, 0.005% asolectin (Avanti) and 0.005% CHAPS (Calbiochem)). The column was washed at 0.3 mL min⁻¹ with buffer A for 2 min, then with 20% buffer B (buffer A + 1 M NaCl) for 7 min, and then complex I was eluted in 35% buffer B for 10 min[4]. Complex I-containing fractions were pooled and concentrated to 100 μL using a 50 kDa MWCO spin column (Sartorius) and the following were then added: 200 μM NADH, 6 μM decylubiquinone (DQ), 0.15% asolectin (20% soy PC, Avanti) and 0.15% CHAPS (Calbiochem) to drive complex I catalysis; 150 μg mL⁻¹ alcohol dehydrogenase from *Saccharomyces cerevisiae* (Sigma) and 1% ethanol to regenerate the NADH from NAD⁺; 100 μg mL⁻¹ alternative oxidase from

*Trypanosoma brucei brucei* (AOX[8]) to regenerate the ubiquinone from ubiquinol; 10 KU mL⁻¹ catalase from *Corynebacterium glutamicum* (Sigma) and 400 U mL⁻¹ superoxide dismutase from bovine erythrocytes (Sigma) to minimise oxidative damage. After ~5 min at 4 °C, 15 μL of the mixture was removed as a control, and the remaining sample added to a glass vial containing sufficient piericidin A (dried from an ethanolic stock solution to avoid addition of solvent) to give 200 μM. Then, each sample was applied to a Superose 6 Increase 5/150 column (GE Healthcare) and eluted in 20 mM Tris-Cl (pH 7.14 at 20 °C), 150 mM NaCl and 0.05% DDM[4]. The concentration of the peak piericidin-bound fraction (at ~1.65 mL) was estimated as 4.1 mg mL⁻¹ using a nanodrop UV–vis spectrophotometer ($\varepsilon_{280} = 0.2$ (mg mL⁻¹)⁻¹). It was found to be 88.7 ± 3.4% inhibited by comparing initial rates of catalysis by the control and inhibited samples in 20 mM Tris-HCl (pH 7.5 at 20 °C), 0.012% asolectin and 0.012% CHAPS, using 5 μM DQ and 10 μg mL⁻¹ AOX, with catalysis initiated by 200 μM NADH and monitored at 340–380 nm ($\varepsilon = 4.81$ mM⁻¹ cm⁻¹). The low DQ concentration was used to minimise competition with the piericidin. The specific activity of the control (measured in 0.075% asolectin, 0.075% CHAPS and 100 μM DQ) was 11.3 ± 0.6 μmol min⁻¹ mg⁻¹.

**CryoEM grid preparation, data acquisition, and processing**. UltrAuFoil® gold grids (0.6/1, Quantifoil Micro Tools GmbH) were glow discharged (20 mA, 60 s), incubated in a solution of 5 mM 11-mercaptoundecyl hexaethyleneglycol (SPT-0011P6, SensoPath Technologies) in ethanol for 2 days in an anaerobic glovebox, then washed with ethanol and air-dried just before use[11]. Using an FEI Vitrobot Mark IV, 2.5 μL of complex I solution (3–5 mg mL⁻¹) was applied to each grid at 4 °C in 100% relative humidity and blotted for 10–12 s at force setting –10, before the grid was frozen by plunging it into liquid ethane. Grids for the piericidin-K2 dataset were imaged using a Gatan K2 detector and GIF Quantum energy filter mounted on an FEI 300 keV Titan Krios microscope with a 100 μm objective aperture and EPU-1.9 software at the UK National Electron Bio-Imaging Centre (eBIC) at Diamond. The energy filter was operated in zero-energy-loss mode with a slit width of 20 eV. Data were imaged at 1.05 Å pixel⁻¹ (magnification 47,600×) with a defocus range −2.2 to −3.4 μm and the autofocus routine run every 10 μm. The dose rate was 5 electrons Å⁻² s⁻¹ with 10 s exposures captured in 25 frames (total dose ~50 electrons Å⁻²). Grids for the piericidin-FIII dataset were imaged using a Falcon III detector on an FEI 300 keV Titan Krios microscope with a 100 μm objective aperture and EPU-1.9 at the Astbury Centre for Structural Molecular Biology, University of Leeds. The detector was operated in counting mode and data were imaged at 1.06 Å pixel⁻¹ (magnification 130,000×) with a defocus range −2.2 to −3.8 μm and the autofocus routine run every 10 μm. The dose rate was 0.64 electrons Å⁻² s⁻¹ with 71.5 s exposures captured in 40 frames (total dose 46 electrons Å⁻²).

All data were processed first using RELION-2.1-patchb1[47]. First, beam-induced movement was corrected for using MotionCor2[48], both with and without dose weighting. CFT estimations were taken from nondose weighted micrographs using GCTF-1.06[49]. Following autopicking and manual curation 60,107 particles were extracted from 1200 K2 micrographs and 76,802 particles from 1454 Falcon III micrographs. The particles were extracted from dose-weighted micrographs and CTF corrected with an amplitude contrast of 0.1 for 2D classification, then CTF parameters were re-estimated with an amplitude contrast of 0.08 thereafter. For the K2 dataset, frames 1–12 were used for movie refinement (total dose 24 electrons Å⁻²) with a running average window of 3, and polished to account for movement and radiation damage using a single frame average for B-factor estimation. For the Falcon III dataset, frames 122 were used with a total dose of 25 electrons Å⁻², with a running average window of 5 and polished using a 3-frame average for B-factor estimation. The particles were then subjected to 3D classification into five classes with angular sampling gradually incremented to 0.9°. For the K2 dataset, the major class containing 27,193 particles was taken forward to the final 3D refinement with solvent flattened FSC curves. For the Falcon III dataset, two classes containing similar numbers of particles at the same resolution were selected and combined in a single class of 36,759 particles for refinement. Subsequently, both datasets plus the previously-described data for the active state of mouse complex I[4] were re-analysed using RELION-3.0. Beam-induced movement was re-analysed using the in-built version of MotionCor2 and the final particles were re-extracted in RELION-3.0, refined, and subjected to Bayesian polishing and CTF refinement, including beam tilt and per-particle astigmatism correction, before the final refinement with solvent flattened FSC curves[50]. 12 frames were used for both K2 datasets, and all 40 frames for the Falcon III dataset. All three maps were post-processed using default parameters using a mask created with the mask creation tool in RELION, and a soft edge of 5, starting from a Molmap command on a pdb model of the piericidin-inhibited enzyme in UCSF Chimera-1.13.1[51]. Map resolution estimates are based on the FSC = 0.143 criterion[52]. Pixel sizes were adjusted to three decimal places; the scaling was determined by using the Chimera "fit in map" command to maximise the correlation between the maps. Example micrographs and 2D classes for all three datasets are shown in Supplementary Fig 2 and local resolution estimates, particle orientation distributions and FSC curves are shown in Supplementary Fig 3.

**Model building, refinement, and validation**. The model for the active mouse (6G2J.PDB[4]) was fitted into the piericidin-K2 map using Chimera[51], then refined against the RELION-sharpened piericidin-K2 map by cycles of manual adjustment

in Coot-0.8.9.1/0.9-pre[53] and real-space refinement in Phenix-1.13-2998 or 1.16-3549[54] with secondary structure restraints. Model building in poorly resolved areas was aided by the unsharpened map and a blurred map. The piericidin A molecule was imported from ChemSpider and restraints generated using AceDRG[55]. The piericidin-K2 final model was then fitted into the active map using Chimera; the piericidin was removed, the B-factors reset to 20, and the model refinement and inspection carried out as above. Model-to-map FSC curves were generated by simulating a map from the model at Nyquist frequency created with Molmap in Chimera[51]. The created map was compared to a masked unfiltered, unsharpened experimental map from RELION using the Xmipp tool in SCIPION-1.2[56]. Final model statistics were produced by Phenix-1.16-3549, MolProbity-4.4[57] (Supplementary Table 1) and EMRinger[58] (scores 3.77 for piericidin K2, 3.33 for active, and 3.06 for the piericidin-K2 model in the piericidin-FIII map). Finally, each model was checked for overfitting by first "shaking" it using Phenix simple dynamics and resetting its B-factors to 20, then refining it against an unsharpened half map filtered to the FSC = 0.143 resolution of the combined map, and comparing the model-to-map FSC curves against each unfiltered, masked half map. Local resolution maps were produced in RELION-3.0 and visualised in Chimera. Small changes to the model (Supplementary Table 1), relative to PDB ID: 6G2J include: identification of a *cis*-proline, present in high resolution structures of the homologous NiFe hydrogenase enzymes such as from *Ralstonia eutropha* (PDB ID: 4IUC)[59], next to cluster N2 (NDUFS7 Pro160); replacement of the ADP bound to NDUFA10 with ATP, with π-stacking between its adenine ring and nearby Phe134; replacement of two PE molecules (M505 and M506 from 6G2J) by a single cardiolipin (N501); improvements to poorly resolved areas such as the N-terminal loops of subunits NDUFS2, NDUFA13, NDUFB7, and NDUFB10.

**EPR measurements**. Piericidin-bound bovine complex I was prepared by combining the standard method for bovine complex I[8,32] with the method for the piericidin-bound mouse enzyme. Starting from 70 mg membrane protein, the sample was concentrated to 500 μL after ion-exchange chromatography. Two hundred and fifty microliter (the nonturnover control) were applied immediately to a 10/300 Superose 6 Increase column; the other 250 μL were divided in two and treated like the mouse preparation to generate a 125 μL "turnover control" without piericidin and a 125 μL sample inhibited with 320 μM piericidin. Both were applied to a 5/150 Superose 6 increase column. All three resulting samples were concentrated, and ∼ 8 μL per sample was placed in a 1.6 mm O.D. Suprasil quartz EPR tube and frozen immediately. Their concentrations were 18 (nonturnover), 13.9 (turnover) and 10 (piericidin-containing) mg mL$^{-1}$ and the maximal activity of the nonturnover control was 16.4 ± 0.2 μmol min$^{-1}$ mg$^{-1}$. All the procedures described were carried out in air (not anaerobically), to match the conditions of cryo-EM grid preparation. EPR measurements were performed using an X/Q-band Bruker Elexsys E580 spectrometer (Bruker BioSpin GmbH, Germany) equipped with a closed cycle cryostat (Cryogenic Ltd. UK) and Bruker Xepr software. Measurements were carried out at X-band (9.5 GHz) using a split-ring resonator module with 2 mm sample access (ER 4118X-MS2)[60]. The temperature and magnetic field were calibrated with an external Cernox thermometer and a Bruker strong pitch sample ($g = 2.0028$) at room temperature. Acquisition times for spectra were approximately 12.5 min per sample. All spectra have been baseline-subtracted using an oxidised complex I sample.

**Preparation and characterisation of proteoliposomes**. Proteoliposomes were produced using complex I from bovine heart and recombinant *Trypanosoma brucei brucei* AOX[8,32]. Briefly, liposomes were formed from 8 mg phosphatidylcholine, 1 mg phosphatidylethanolamine and 1 mg cardiolipin (bovine heart extracts from Avanti Polar Lipids) together with varying amounts of Q$_{10}$ (Sigma-Aldrich) in 10 mM Tris-SO$_4$ (pH 7.5 at 4 °C) and 50 mM KCl. Following extrusion with a 0.1 μm track etched membrane, they were partly solubilized with 1.5% octyl-glucoside (Anatrace). AOX (0.2 mg) and then complex I (0.2 mg) were added for reconstitution. Aliquots of Biobeads (Bio-Rad) were added hourly over 4 h to remove the detergent then the proteoliposomes collected by centrifugation, resuspended, and flash frozen for storage at −80 °C. Total protein concentrations were quantified using the amido black assay, and the concentration and orientation of complex I using the NADH:APAD$^+$ activity assay[8,32]. Total phospholipid contents were determined as follows[8,32]. Hundred microliter of sample or KH$_2$PO$_4$ as standard, 50 μL of methanol and 30 μL of 390 mM Mg(NO$_3$)$_2$ in ethanol were heated in boiling tubes over a roaring blue flame until no further brown fumes were formed to leave a white residue. After 5 min, 0.3 mL of 500 mM HCl was added, and the tubes were lightly stoppered, incubated at 99 °C for 15 min, cooled, and then 0.7 mL of an aqueous solution of 114 mM ascorbic acid, 2.72 mM (NH$_4$)$_6$Mo$_7$O$_{24}$ and 400 mM H$_2$SO$_4$ was added, and the tubes incubated at 37 °C for 1 h. The absorbances were measured at 820 nm. One milligram of phospholipid was assumed to occupy a volume of 1 μL[8,32]. Q$_{10}$ was quantified by HPLC. Five to ten microliter of sample were solubilised in 190 μL HPLC grade ethanol, then 50 μL loaded onto a Nucleosil 100-5C18 (Hichrom) column maintained at 30 °C on an Agilent 1100 series HPLC system. The mobile phase (run at 800 μL min$^{-1}$) contained 70% ethanol, 30% methanol, 0.7% NaClO$_4$, 0.07% HClO$_4$[8] with elution monitored using a Dionex Ultimate 3000 RS electrochemical detector. A conditioning electrode (6020RS omni Coulometric cell) placed before the sample injector was set to +1000 mV and the dual electrodes of the detecting Coulometric cell (6011RS Ultra Analytical cell)

were set to −500 and +450 mV. The ubiquinone content was calculated from its +450 mV-peak area, by comparison to a set of standards. Ubiquinone concentrations are expressed as nmol of Q$_{10}$ per mg of phospholipids, which equates to mM concentration units.

**Piericidin inhibition kinetics of complex I-AOX proteoliposomes**. Catalytic activity assays were conducted at 32 °C in 10 mM Tris-SO$_4$ (pH 7.5 at 32 °C) and 50 mM KCl in 96-well plates using a Molecular Devices Spectramax 384 plus platereader with Softmax Pro 5.4.42.1. Piericidin was added in ethanol and catalysis initiated by addition of 200 μM NADH and monitored at 340 and 380 nm ($\varepsilon_{340-380}$ = 4.81 mM$^{-1}$ cm$^{-1}$). Linear rates were measured, typically after a 100 s period of equilibration and/or activation. Inhibitor insensitive rates (recorded using 20 nM piericidin) were subtracted. The data are presented as mean values with standard deviations propagated from the quantifications of enzyme, sidedness, membrane volume, quinone content as well as the activity, which was performed in triplicate. The data were modelled using the ordinary differential equation solver, ode15s, in MATLAB (Mathworks, R2018a) with all the reverse rate constants ($k_{-1}$, $k_{-3}$, etc.) set to 1. $k_4$ was set to 1000 s$^{-1}$ to ensure reoxidation of ubiquinol by AOX was not rate limiting[8]. Rate calculations were terminated once they reached steady state (judged as a change in enzyme-substrate complex concentration of <1 fM s$^{-1}$ upon each simulated timestep). The overall rate of NADH oxidation was calculated by multiplying the steady-state concentration of enzyme-substrate complex with the turnover rate ($k_2$). Fitting was initiated repeatedly with a random set of values for the variable parameters, and targeted on the parameter combination with the smallest error between the calculated and measured rates; the data were modelled simultaneously over all Q$_{10}$ and piericidin concentrations. The piericidin log$_{10}P$ value (4.73) was calculated in ChemSpider (http://www.chemspider.com). For most final models, bootstrapping of the residuals ($n = 1000$) was conducted using the triplicate activity measurements (rather than the means) in order to derive fitting statistics (mean, median, and 95% confidence intervals) of model parameters. For poorly-fitting models, uncertainties in fit parameters are the standard errors of fitting from the nonlinear regression.

**Molecular simulations**. All-atom classical molecular dynamics (MD) simulations (see Supplementary Table 4) were performed using our resolved piericidin-K2 structure of mouse complex I. Chains A, B, C, D (residues 34–430), H, I, J, K, P, a, b, e, r, W, X (residues 1–150), and Z were embedded in a POPC membrane and solvated with TIP3P water and 100 mM NaCl. The entire system comprised ∼396,000 atoms. Force field parameters for the cofactors were derived from DFT models[61]. Parameters for piericidin were estimated initially using CGenFF[62], and refined at B3LYP-D3/def2-TZVP level[63]. Protonation states of key titratable amino acids were determined by Poisson–Boltzmann electrostatic (PBE) calculations[64] and long-range electrostatics treated by the Particle Mesh Ewald method. All simulations were performed at 1 atm, 310 K, with a 2 fs integration timestep. After initial minimisation, the system was heated up to the final temperature using a set of harmonic restraints (2 kcal mol$^{-1}$ Å$^{-2}$) on the entire protein structure and cofactors (0.5 ns), then in a second step restrains were applied only to the backbone atoms (1 ns). Simulations were performed using NAMD 2.9/2.13[65] and analysed using VMD[66].

The initial structure of piericidin near Tyr108 was based on the cryoEM model (simulation A1–A3, A5–A6), or docked in de novo (simulation A4) using the cryoEM density. The structure of the piericidin in the lower part of the cavity (simulation A1, A2, and A4) was based on the local quinone-binding substrates described previously, that correspond to local binding minima identified in simulations of *T. thermophilus* complex I[34]. The second binding site was also probed by molecular docking calculations. Although some docked models could be identified with qualitative resemblance to the MD poses, the best scoring models were qualitatively incorrect, and the approach was therefore not employed for further exploration of the site. Water clustering analysis was performed on the last 40 ns (2000 frames) of simulation A1 using the WATCLUST method[67], with standard clustering values, and a 10% threshold for cluster recognition. Water molecules ∼3 Å from residues NDUFS2 Asp107, Thr156, Lys371, Asp422, and Val424 were considered in the clustering analysis.

All setups were optimised using the molecular dynamics flexible fitting (MDFF) procedure[27]. MDFF simulations of piericidin were carried out using the inner core of the quinone-binding cavity (subunits ND1, NDUFS2, and NDUFS7), and imposing secondary structure restraints on the backbone atoms of the protein, plus a distance restraint of 1.7 Å between the Tyr108 hydroxyl and piericidin 1′ carbonyl. After initial minimisation, the system was equilibrated at 310 K for 1 ns, followed by minimisation with high restraints on the experimental density. Simulations based on the cryoEM model were subject only to the latter. Energy decomposition analyses comprised all residues within 5 Å, and included interactions with protein backbone atoms.

Coarse-grained molecular dynamics (CGMD) simulations (see Supplementary Table 4) were performed using the MARTINI 2.2 force field[68]. In total, 10 × 10 μs simulations using a 20 fs timestep, a semi-isotropic Parrinello–Rahman barostat with a coupling constant of $\tau_p = 12.0$ ps, and a compressibility of $\chi = 3.0 \times 10^{-4}$ bar$^{-1}$ were carried out. The simulation temperature was set to $T = 310$ K with a thermostat coupling constant of $\tau_t = 1.0$ ps. Nonbonded interactions were treated using a cutoff distance of 11 Å and $\varepsilon = 15$. Piericidin was placed in different

starting positions across the quinone channel. The CGMD simulations performed using Gromacs 2016.3[69].

To obtain a parameter-free estimation of binding poses in the active site, QM/MM MD simulations (see Supplementary Table 4) were performed with the piericidin, Tyr108, His59, Thr156, and Asp160 in the QM region treated at the B3LYP-D3/def2-SVP level of theory[63]. Protein residues within 10 Å of the piericidin were allowed to move during the dynamics. The total QM/MM system was trimmed to include 9100 atoms. The classical region was modelled at the CHARMM36 level of theory in combination with in-house DFT parameters for the cofactors (see above). All QM/MM simulations were performed using the CHARMM c38b, TURBOMOLE 6.6–7.3 and the CHARMM/TURBOMOLE interface[70].

A complete list of all simulations performed are shown in Supplementary Table 4.

**Reporting summary**. Further information on research design is available in the Nature Research Reporting Summary linked to this article.

## Data availability

Data accession codes: EMD-11424, PDB ID: 6ZTQ (piericidin-bound structure), EMD-11425 (piericidin-bound map from the FIII detector), EMD-11377, PDB ID: 6ZR2 (active state structure). Other data supporting the findings of this manuscript are available from the corresponding authors upon reasonable request.

## Code availability

The Matlab algorithms used for kinetics modelling are available from the corresponding authors upon request.

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

## Acknowledgements

This work was supported by The Medical Research Council (MC_U105663141 and MC_UU_00015/2 to J.H.), the European Research Council (ERC) under the European Union's Horizon 2020 research and innovation programme/grant agreement no. 715311 (to V.R.I.K.), the Knut and Alice Wallenberg Foundation (to V.R.I.K.) and the Royal Society (RGS-R1-191215 to M.M.R.). CryoEM data were recorded at the Astbury Biostructure Laboratory, University of Leeds (funded by the University of Leeds and Wellcome Trust 108466/Z/15/Z) and at the UK National Electron Bio-Imaging Centre (eBIC) at the Diamond Light Source, proposal EM16309, funded by the Wellcome Trust, MRC and BBSRC. We thank Steven Muench (Leeds) and Corey Heckel (Diamond) for their support with data collection. Computational resources were provided by the PRACE (project nr: 2018194738), awarding access to MareNostrum at the Barcelona Supercomputing Centre (BSC), Spain and by the Swedish National Infrastructure for Computing (SNIC, 2019/2-3). Queen Mary University of London is gratefully acknowledged for EPR measurement time.

## Author contributions

H.R.B., J.N.B., and A.N.A.A. prepared complex I and cryoEM grids and processed cryoEM data; H.R.B. and J.N.B. collected cryoEM data and built the models; H.R.B., J.G.F., J.N.B., V.R.I.K., and J.H. analysed and interpreted the models; A.D.L., A.J., A.P.G. H., and V.R.I.K. performed and analysed molecular simulations; O.D.J. prepared proteoliposomes and performed kinetic experiments and analyses; J.G.F. performed kinetic modelling; J.J.W. and M.M.R. performed and analysed EPR spectroscopy; V.R.I.K. and J.H. directed the project; H.R.B., J.G.F., V.R.I.K., and J.H. wrote the manuscript with input from all authors.

## Competing interests

The authors declare no competing interests.
