## [Peer Review File · Nature Communications]

REVIEWER COMMENTS

Reviewer #1 (Remarks to the Author):

Applying a combination of cryoEM structure analysis, molecular modeling, EPR and enzyme kinetics Bridges et al. present the first comprehensive analysis of inhibitor binding to proton pumping complex I at the atomic level. This study represents an important step forward towards our understanding of the still enigmatic mechanism of energy conversion by this respiratory chain enzyme. The cryoEM structure is of excellent quality and it is remarkable that a resolution of 3.0 Å was obtained collecting far less than 100k particles. The molecular modeling analysis of the structural environment of the bound inhibitor is very valuable and adds significant additional insights. The standard and thoroughness of the biochemical experiments is equally high and all results are very well presented and easy to follow.

I have just one remark concerning the analysis of the enzyme kinetics used to argue in favor of simultaneous binding of two inhibitor molecules to the complex. In itself the argument is conclusive and gains some support from the densities observed in the lower parts of the substrate binding channel and molecular modeling. However, the analysis is based on the Michaelis-Menten model of steady-state kinetics that has some limitations. Specifically, it is built on the law of mass action and thus assumes free diffusion of the substrates and products to and from the active site of the enzyme. In other words, the model assumes that the reaction is limited by product formation and not substrate access. Considering the unusually long and narrow substrate access channel it seems quite possible that complex I turnover is limited by the rate of substrate access rather than ubiquinone reduction. Thus, the Michaelis-Menten model may not be applicable rendering conclusions on competitive or non-competitive inhibition impossible. This potential limitation of the kinetic analysis presented should be considered carefully and at least be discussed.

Reviewer #2 (Remarks to the Author):

The work entitled "Structure of inhibitor-bound mammalian complex I" presents a very interesting and relevant work that reveals new insights into mammalian complex I function, particularly in relation to the action mechanism of a "substrate-like" inhibitor, pieridin. The works present high resolution CryoEM results, combined with reaction kinetic/inhibition studies, and molecular simulations (classical, hybrid QM/MM and Coarse Grained) data.

The work is well done and presented, I have only a few minor points noted below that need to be addressed before publication.

1) Concerning the hybrid QM/MM simulations, authors state that "Hybrid quantum/classical (QM/MM) simulations, in which the binding site was modeled using density functional theory (DFT) polarized by a classical model for the surrounding protein, also revealed the hydrogen-bonded and van der Waals binding poses." this is the only result presented from the QM/MM simulations, which contributes little to the issue at hand. Given its very minor role, I would suggest to remove QM/MM data. QM/MM data should be included when either reactive (bond forming/breaking or electron transfer) processes are modeled. To analyze interactions classical MD simulations is fine and can be even better than QM/MM.

2) The authors state that for binding site 1. "Piercidin was modeled into the density or docked in using MD flexible fitting (MDFF) with the density as a biasing potential." For the second site, authors just state "we performed MD simulations with a second piercidin added", but there is no hint as to how they built the corresponding structure, moreover considering that they themselves stated that CryoEM density did not allow to identify the second site. Also, I (and possibly wide audience) will not be familiar with the MDFF methodology (no clear reference added.) Therefore, and given that many results of the current work is structures of complex I bound to one/two

piercidins I would suggest authors to test whether any docking program, is able to yield the proposed binding sites/conformations. Confirmation of the piercidin binding sites and poses using molecular docking will definitely add support to the present work.

3) The analysis of both piercidin binding sites is unbalanced. While the first is analyzed in detail (Figure 3), including estimated interaction energies, scarce (or none) interaction data is presented for site 2. I would suggest the authors to perform a comparative analysis of both sites, in terms of interaction and possible binding free energy using classical methods.

4) Authors state that: "kinetic data suggest the second inhibitor binds substantially more tightly than the first, ... with a 10,000x tighter binding of the second piercidin than the first." Although I agree that this is what kinetic results show. The structural interpretation presented later (defining the nature of the two sites) leads to a different interpretation/conclusion that I think should be discussed in the corresponding section. If the second piercidin binds where proposed i.e distal to the first one and closer to the entry of the binding pocket, and once proposed comparative analysis is performed, it will possibly show that possibly it displays a smaller binding affinity (otherwise this is the molecule that should be evidenced in the CryoEM map). Smaller affinity is the opposite as suggested by the kinetics, why???. Well, the most parsimonious explanation is that binding of the second piercidin significantly increases first piercidin affinity, since to leave the binding site, the second piercidin has to be released since its blocking the way.

5) Authors state that: "Interestingly, weak densities observed between the Thr156 hydroxy, the piercidin methoxy groups, and NDUFS2 Lys371 may arise from bound water molecules and reflect a wider hydrogen-bonded network for ubiquinone protonation/reduction. However, we are not sufficiently confident to model water networks at the current resolution." Authors, could you use explicit water MD simulation data and characterize the water networks using any of the available Water Site (WS) detection algorithms ([10.1093/bioinformatics/btv411](https://doi.org/10.1093/bioinformatics/btv411) for example).

Minor issues: in the methods section, authors should clearly state the length of the classical MD simulations.

Reviewer #3 (Remarks to the Author):

Bridges et al. studied the binding of the inhibitor piercidin A to the mouse respiratory complex I. Structural, kinetic and simulation data convincingly show that two inhibitor molecules bind in the substrate channel of complex I. Considering the importance of complex I as a central enzyme in metabolism, the authors provide a clear and interesting investigation of general interest.

Some specific points in the manuscript the authors may want to consider:

line 70-71: The authors should state when the complex enters the deactivated state and how it can be reactivated.

line 121: I wondered if the authors tried to combine both data sets at the end and refine them together?

line 130-132, Figure 1a: In Figure 1a the overlay of both densities does not work as it is not possible to distinguish them. Only the carved densities of the two data sets for the inhibitor are shown. This does not really allow to appreciate how well the density stands up against the noise. The authors should show the density of the inhibitor (without carving) in the context of the surrounding residues, perhaps at two different thresholds and as a slice for clarity? This would also allow the reader to assess the densities for potential water molecules (as mentioned in line 158-161). Since this is the main point of the paper the authors could also present a difference map of the region.

line 201-204: The authors write here something different than they present in Figure 4a. In the text they mention a turnover control without piercidin while in the Figure 4a they show a spectrum of a piercidin sample as prepared. Please clarify. In general, I find the assumption that N2 is

reduced in the piericidin-bound structure weakly supported. First, the EPR experiment was done (understandably!) on the bovine complex I. But we don't know how the redox potential of N2 in the mouse complex I compares to that of bovine. In addition, freezing the grids is a different freezing from EPR samples in the tube. The enzyme is after blotting exposed to the air-water interface which may lead to faster reoxidation.

line 452-466: Why were baseline subtractions with the oxidised complex I done? Or what is the difference between the non-turnover sample and the oxidised sample? Were the SEC run and concentration step done under anaerobic conditions?

SI: Please, show some standard EM characterisation: example of a micrograph, typical 2D classes, FSC, angular distribution.

Reviewer #4 (Remarks to the Author):

The paper by Bridges et al. is a very careful study of the specific inhibitor, piericidin, binding to complex I. All used different protocols are very well described; their number is impressive. The structure of complex I with bound piericidin was resolved, and it was shown that the inhibitor is bound in the quinone-binding site as expected due to the high similarity of ubiquinone and piericidin molecules.

Also, the authors claim that the second piericidin-binding site occurs close to the bottom of the quinone-binding channel, and the conclusion of two-site competitive inhibition was made. However, this conclusion probably should be more tentative since the inhibitor-enzyme interaction could be more complicated than described with simple Michaelis-Menten curves, and the number of experimental points for quinone K_m determination should be increased for a precise analysis. The Discussion provides a broad description of the literature data on inhibitor binding however, the analysis of the data obtained seems insufficient. The finding of two piericidin-binding sites arises several questions. The model of two-site competitive binding predicts 10 000 folds difference in the binding tightness. One site is clearly at the top of the ubiquinone binding channel in N2 cluster proximity. If this site is tight-binding, it would be important to compare piericidin and the ubiquinone binding. Since the structures of these compounds are very similar, what particularly makes piericidin to be bound that tight in contrast to ubiquinone? Or could it be that the tight-binding occurs at the second site located partially in ND1? The very interesting and important suggestion that piericidin binding to the second site could be relevant to proton translocation (Discussion, last line) needs some elucidation: how particularly the authors imagine this relevance. The choice of quoting is embarrassing. P.2, 52-57. These lines make the impression that the architecture of the electron transfer chain was not known before publishing the papers cited in the manuscript. However, the position of redox groups (Sazanov's studies) and the identification of the terminal cluster N2 (Ohnishi's studies) in the bacterial complex I were known for a long time. Because core subunits of eu- and prokaryotic complex I are highly similar and the structure of the electron transfer chain is almost identical the earliest papers should be cited here. In the discussion of inhibitor binding sites in complex I, the studies of Miyoshi's group, who made a huge contribution to this issue, are not even mentioned. The paper by Degli-Esposti, who suggested the second piericidin-binding site partially in ND1 (exactly at the same position as in the manuscript), is mentioned but not in this context.

Minor comments:

p.4, l.140-141. It is not clear whether piericidin binds to the active state of complex I, or it could bind the de-active state but does not prevent conversion the de-active-to-active state. On the other hand, if before piericidin binding the enzyme clearly was in the active state, how the authors know that after inhibitor binding complex I did not convert to the inactive state?

P.6. l.198-218 and Fig. 4. The description of EPR experiment is not clear. The EPR spectrum of the "turnover" sample before and after following thawing and NADH addition should be shown in Fig.4 for comparison with the "inhibited" sample. Were the "turnover" and "inhibited" samples prepared

with NADH, Q10 and AOX? Both samples were applied to the Superose column; therefore they should be depleted on NADH, and oxidized by AOX in the presence of oxygen in a way dependent on quinone reductase activity. A comparison of such spectra will show the retention of electron on the N2 cluster due to piericidin binding.

Reviewer #1 (Remarks to the Author):

Applying a combination of cryoEM structure analysis, molecular modeling, EPR and enzyme kinetics Bridges et al. present the first comprehensive analysis of inhibitor binding to proton pumping complex I at the atomic level. This study represents an important step forward towards our understanding of the still enigmatic mechanism of energy conversion by this respiratory chain enzyme. The cryoEM structure is of excellent quality and it is remarkable that a resolution of 3.0 Å was obtained collecting far less than 100k particles. The molecular modeling analysis of the structural environment of the bound inhibitor is very valuable and adds significant additional insights. The standard and thoroughness of the biochemical experiments is equally high and all results are very well presented and easy to follow.

We thank the reviewer for these positive remarks.

I have just one remark concerning the analysis of the enzyme kinetics used to argue in favor of simultaneous binding of two inhibitor molecules to the complex. In itself the argument is conclusive and gains some support from the densities observed in the lower parts of the substrate binding channel and molecular modeling. However, the analysis is based on the Michaelis-Menten model of steady-state kinetics that has some limitations. Specifically, it is built on the law of mass action and thus assumes free diffusion of the substrates and products to and from the active site of the enzyme. In other words, the model assumes that the reaction is limited by product formation and not substrate access. Considering the unusually long and narrow substrate access channel it seems quite possible that complex I turnover is limited by the rate of substrate access rather than ubiquinone reduction. Thus, the Michaelis-Menten model may not be applicable rendering conclusions on competitive or non-competitive inhibition impossible. This potential limitation of the kinetic analysis presented should be considered carefully and at least be discussed.

First, we agree that the Michaelis-Menten (MM) model is an over-simplification of the mechanism of complex I catalysis, and that quinone binding/dissociation are probably complex multi-step processes (see, for example, Fedor, J.G., et al. (2017) PNAS and Teixeira et al. (2019) Biochem. Biophys. Acta). However, the actual kinetic modelling done here differs from classical MM kinetics in two important ways. First, we allowed for all the forward kinetic parameters to vary independently, while the reverse rates were set to 1. So, we do not assume that the rate of quinone binding (k_1) greatly exceeds the rates of quinone reduction and quinol release (k_2) – either k_1 or k_2 may be rate limiting. For evaluating our models, ordinary differential equations utilizing the individual rate constants were used, and we defined K_M as $(k_{-1}+k_2)/k_1$ (technically, this is the Briggs-Haldane model, whereas in the Michaelis-Menten model $K_M \rightarrow k_{-1}/k_1$). Second, where traditional MM analysis assumes negligible substrate depletion during turnover, we account for substrate and inhibitor depletion as well as piericidin partitioning between the aqueous and membrane phases. We have now amended our text on page 7 to make the lack of assumptions about the rate limiting step clearer: “Importantly, piericidin is a tight-binding inhibitor, and the pool of ubiquinone/ubiquinol is small, so depletion of their free concentrations upon binding were considered. Although this precluded simple analytical solution of the rate equations and required kinetic rates to be calculated numerically, the numerical approach also enabled us to test essentially any reasonable model without having to make simplifying assumptions, such as about the identity of the rate determining step in complex I catalysis.”

Second, in an attempt to capture the complexity of quinone/ol association/dissociation by multi-step processes we considered and tested many more models than presented in the paper, including multiple Enzyme-Substrate (and Enzyme-Product) states. However, we found that these models, when they did not include two binding sites for the inhibitor, did not account for the data – they only introduced ridiculously wide ranges to the fitting parameters (owing to their co-variance). We therefore believe that our kinetic analysis holds and the conclusions drawn from it are valid, although we are aware that it does not have capability to tease out the finer kinetic details of turnover and inhibition.

Reviewer #2 (Remarks to the Author):

The work entitled “Structure of inhibitor-bound mammalian complex I” presents a very interesting and relevant work that reveals new insights into mammalian complex I function, particularly in relation to the

action mechanism of a “substrate-like” inhibitor, pieridin. The works present high resolution CryoEM results, combined with reaction kinetic/inhibition studies, and molecular simulations (classical, hybrid QM/MM and Coarse Grained) data. The work is well done and presented, I have only a few minor points noted below that need to be addressed before publication.

We thank the reviewer for these positive remarks.

1) Concerning the hybrid QM/MM simulations, authors state that “Hybrid quantum/classical (QM/MM) simulations, in which the binding site was modeled using density functional theory (DFT) polarized by a classical model for the surrounding protein, also revealed the hydrogen-bonded and van der Waals binding poses.” this is the only result presented from the QM/MM simulations, which contributes little to the issue at hand. Given its very minor role, I would suggest to remove QM/MM data. QM/MM data should be included when either reactive (bond forming/breaking or electron transfer) processes are modeled. To analyze interactions classical MD simulations is fine and can be even better than QM/MM.

We thank the reviewer for this suggestion, but we respectfully disagree that the QM/MM does not provide additional insight into the binding pose. As the reviewer correctly points out, QM/MM simulations are powerful for studying reactivity, but importantly, they also predict molecular structures without the bias of pre-parametrized geometry, which always underlies classical force field-based simulations. Given the discrepancy between distances in the cryoEM and classical MD for the key Tyr108-Q10 interaction, our QM/MM simulations validate the binding geometry predicted by the classical simulations. We therefore wish to keep the QM/MM data as an important method validation.

2) The authors state that for binding site 1. “Piercidin was modeled into the density or docked in using MD flexible fitting (MDFF) with the density as a biasing potential. “For the second site, authors just state “we performed MD simulations with a second piercidin added”, but there is no hint as to how they built the corresponding structure, moreover considering that they themselves stated that CryoEM density did not allow to identify the second site. Also, I (and possibly wide audience) will not be familiar with the MDFF methodology (no clear reference added.) Therefore, and given that many results of the current work is structures of complex I bound to one/two piercidins I would suggest authors to test whether any docking program, is able to yield the proposed binding sites/conformations. Confirmation of the piercidin binding sites and poses using molecular docking will definitely add support to the present work.

We thank the reviewer for these excellent suggestions. We have now added a reference to the MDFF method on page 5 (Flexible fitting of atomic structures into electron microscopy maps using molecular dynamics. Leonardo G. Trabuco, Elizabeth Villa, Kakoli Mitra, Joachim Frank, and Klaus Schulten. Structure, 16, 673-683, 2008). Following the reviewer’s advice, we also performed docking calculations. We observe that a second piercidin that resembles (Δ RMSD \sim 2.3 Å for heavy atoms) the piercidin from the MD models is predicted amongst the poses with the 10 best binding scores. In these models, the tail of the docked piercidin points toward the entrance to the Q tunnel, consistent with the cryo-EM density, but only when another piercidin molecule is modelled in the primary binding site (i.e. when the ‘primary’ piercidin is absent the docking simulations fail to predict structures with the piercidin tail pointing toward the tunnel exit). However, the best scoring models are (unfortunately) qualitatively incorrect – and therefore we have chosen not to present the new data in our revised manuscript. The multiple binding poses predicted may reflect the sparse density observed, the co-operative interaction between the two molecules, and also reflect/explain our difficulties in identifying a well-defined unique binding site.

The best scoring docking models from left to right: In the top three models, the piercidin tail and headgroup are predicted in qualitatively incorrect binding poses.

We have added a short section to our methods section accordingly (page 13): “The second binding site was also probed by molecular docking calculations. Although some docked models could be identified with qualitative resemblance to the MD poses, the best scoring models were qualitatively incorrect, and the approach was therefore not employed for further exploration of the site.”

3) The analysis of both piericidin binding sites is unbalanced. While the first is analyzed in detail (Figure 3), including estimated interaction energies, scarce (or none) interaction data is presented for site 2. I would suggest the authors to perform a comparative analysis of both sites, in terms of interaction and possible binding free energy using classical methods.

We decided to focus less on site 2 due to the complex definition of this dynamic and flexible site, and the relative paucity of experimental evidence for comparison with the simulation data. We note that an accurate calculation of binding free energies requires a complete conformational sampling, which is beyond the scope of the current manuscript. We have therefore limited our discussion on the second site to a qualitative level. Following the reviewer’s advice, we have now provided (in Supplementary Figure 5), in addition to the previously reported interaction energies for the two piericidin headgroups, also the interaction energies for the piericidin tail, in order to guide future experiments.

4) Authors state that: “kinetic data suggest the second inhibitor binds substantially more tightly than the first, ... with a 10,000x tighter binding of the second piericidin than the first.” Although I agree that this is what kinetic results show. The structural interpretation presented later (defining the nature of the two sites) leads to a different interpretation/conclusion that I think should be discussed in the corresponding section. If the second piericidin binds where proposed i.e distal to the first one and closer to the entry of the binding pocket, and once proposed comparative analysis is performed, it will possibly show that possibly it displays a smaller binding affinity (otherwise this is the molecule that should be evidenced in the CryoEM map). Smaller affinity is the opposite as suggested by the kinetics, why???

The reviewer has raised some interesting and challenging questions, and indeed we spent considerable time and effort agonising over exactly these issues. First, we have noted the differences (described in the literature) between the detergent-bound and phospholipid-bound enzyme – one and two bound piericidins, respectively (page 8). Consistent with our data the literature suggests that the second piericidin is bound much more weakly in the detergent-bound (delipidated) enzyme, consistent with its location at the entrance to the binding tunnel, and this may explain why we fail to observe clear density for it in the cryoEM analyses.

Turning to an explanation for the proposed higher affinity of the second molecule in the phospholipid state, we have now expanded our discussion in the text to clarify how the many subsites of the dispersed second binding are all expected to contribute to the binding affinity: an entropic stabilization of the set of subsites to provide a higher affinity for the site as a whole. We also note that interactions (particularly the potential hydrogen bond) between the two piericidins contribute to the apparent cooperativity. Indeed, the 5 kcal mol⁻¹ interaction energy between piericidins 1 and 2, comparable to a strong hydrogen bond, accounts for a 3000-fold enhanced K_d for piericidin 2 when piericidin 1 is present relative to when it is absent. We have added the following to our text on page 8 accordingly: “However, the distal piericidin does not adopt a single, well-defined binding pose, and its fluctuations in the MD simulations, together with the unclear density, are better explained by a broad substrate recognition region, rather than a unique, well-defined distal site. The broad recognition region may lead to an entropic stabilization for the set of subsites to provide a higher affinity for the site as a whole, whereas the interaction between the two piericidins may contribute to the cooperativity, explaining the tighter apparent binding of the second piericidin relative to the first.”

5) Authors state that: “Interestingly, weak densities observed between the Thr156 hydroxy, the piericidin methoxy groups, and NDUFS2 Lys371 may arise from bound water molecules and reflect a wider hydrogen-bonded network for ubiquinone protonation/reduction. However, we are not sufficiently confident to model water networks at the current resolution.” Authors, could you use explicit water MD simulation data and characterize the water networks using any of the available Water Site (WS) detection algorithms (10.1093/bioinformatics/btv411 for example).

We thank the reviewer for this suggestion. It is correct that we are not confident to assign bound water molecules in our model, however the new panels added to Figure 2 now illustrate the densities we refer to so that the reader can see them and evaluate them. We have also adopted the suggestion to use the

WATCLUST algorithm to predict the waters in this region of the protein, in the pocket between Lys371 and Thr156. We have thus included the additional Supplementary Figure 5b to show the results of the water clustering analysis, and have briefly described the results in the text on page 6: “Commensurate with the additional densities observed in the cryoEM analyses, water clustering analyses (see Supplementary Fig. 5b and Methods) suggest that this region could form a stable water binding site, stabilised by the nearby conserved residues NDUFS2 Asp107 and Asp422.” Also, in the methods section on page 13: “Water clustering analysis was performed on the last 40 ns (2000 frames) of simulation A1 using the WATCLUST method⁶⁵, with standard clustering values, and a 10% threshold for cluster recognition. Water molecules ~ 3 Å from residues NDUFS2 Asp107, Thr156, Lys371, Asp422, Val424 were considered in the clustering analysis.”

Minor issues: in the methods section, authors should clearly state the length of the classical MD simulations.

The simulations are listed in Supplementary Table 4, which is now clearly referred to in the Methods section.

Reviewer #3 (Remarks to the Author):

Bridges et al. studied the binding of the inhibitor piericidin A to the mouse respiratory complex I. Structural, kinetic and simulation data convincingly show that two inhibitor molecules bind in the substrate channel of complex I. Considering the importance of complex I as a central enzyme in metabolism, the authors provide a clear and interesting investigation of general interest.

We thank the reviewer for these positive remarks.

Some specific points in the manuscript the authors may want to consider:

line 70-71: The authors should state when the complex enters the deactivated state and how it can be reactivated.

We have extended our explanation to read: “The structures of the mammalian enzyme in both its active and deactive states depict how these loops become disordered when complex I converts from its ready-to-catalyse ‘active’ state to its ‘deactive’ state. The deactive state is a pronounced resting state that forms spontaneously at physiological temperatures in the absence of turnover, and requires reactivation by both NADH and ubiquinone in order to re-enter the catalytic cycle.”

line 121: I wondered if the authors tried to combine both data sets at the end and refine them together?

We did indeed try this approach. We achieved a marginal increase in resolution, but with no visual improvement in the map quality, and the piericidin density in the combined map was less well-defined than in the two individual maps. We therefore chose not to include this analysis in our manuscript. We noted an unusually high degree of beamtilt in the Falcon3 data suggesting that a microscope alignment issue could be contributing to the disappointing outcome from dataset merging.

line 130-132, Figure 1a: In Figure 1a the overlay of both densities does not work as it is not possible to distinguish them. Only the carved densities of the two data sets for the inhibitor are shown. This does not really allow to appreciate how well the density stands up against the noise. The authors should show the density of the inhibitor (without carving) in the context of the surrounding residues, perhaps at two different thresholds and as a slice for clarity? This would also allow the reader to assess the densities for potential water molecules (as mentioned in line 158-161). Since this is the main point of the paper the authors could also present a difference map of the region.

We agree, and we thank the reviewer for this helpful suggestion. To allow the density of the piericidin molecule to be evaluated more fully we have included new panels in Figure 2 to show the density of the piericidin and surrounding protein in both piericidin-containing maps, plus the difference map between the two piericidin maps and the active model in this area.

line 201-204: The authors write here something different than they present in Figure 4a. In the text they mention a turnover control without piericidin while in the Figure 4a they show a spectrum of a piericidin sample as prepared. Please clarify.

We apologise for the confusion and thank the reviewer for bringing it to our attention. An additional panel with the turnover control data has been added to Figure 4 and the text in the paragraph headed ‘The environment of cluster N2’ has been amended to have consistent terminology with the figure and the experimental section. We hope that everything is now clear.

In general, I find the assumption that N2 is reduced in the piericidin-bound structure weakly supported. First, the EPR experiment was done (understandably!) on the bovine complex I. But we don’t know how the redox potential of N2 in the mouse complex I compares to that of bovine. In addition, freezing the grids is a different freezing from EPR samples in the tube. The enzyme is after blotting exposed to the air-water interface which may lead to faster reoxidation.

First, the EPR spectra of all eukaryotic complexes I studied so far are very similar – for example, the N2 spectrum in bovine complex I and complex I from the yeast *Yarrowia lipolytica* match closely and they both exhibit N2 reduction potentials around -140 mV at pH 7, despite their phylogenetic distance. Furthermore, the environment of N2 is highly conserved. The second co-ordination sphere around the cluster (the residues within 8 Å) is identical between the bovine and mouse enzymes. We therefore are confident in expecting the reduction potential of N2 in mouse complex I to match that in bovine complex I.

Second, the samples for both cryo-EM and EPR analyses were exposed to air throughout the preparation, including for ~1 hour after the NADH had been removed by gel filtration. Therefore, they are already fully aerobic and so the greater extent of the air/sample interface during the short cryo-EM grid preparation procedure will not affect the protein oxidation state. For clarity we now state in the EPR methods section that “All the procedures described were carried out in air (not anaerobically, to match the conditions of cryo-EM grid preparation).”

We hope that the reviewer will now agree that our assumption of a matching N2 oxidation state in the two samples is reasonable.

line 452-466: Why were baseline subtractions with the oxidised complex I done?

Or what is the difference between the non-turnover sample and the oxidised sample? Were the SEC run and concentration step done under anaerobic conditions?

Complex I prepared without turnover is oxidised and it is well established that it does not produce any EPR signals in the absence of reducing agents. A separate sample of oxidised complex I was used as the baseline sample because it has a matching composition to the samples being measured and can be used to remove meaningless (highly reproducible) signals that arise from contamination of the instrument EPR cavity. We performed this procedure as good practice (it is standard amongst EPR spectroscopists as cavities do get contaminated in the long term) although, in our case, the adjustments to the spectra were minimal.

Essentially, there is no difference between the oxidised ‘standard’ sample and the non-turnover sample, except that the non-turnover sample is specific to the current experiment.

None of the procedures were carried out anaerobically, for either the EPR or the cryo-EM preparations – they were all performed in air.

SI: Please, show some standard EM characterisation: example of a micrograph, typical 2D classes, FSC, angular distribution.

We have added new panels to Figure S2 (now S3) to show the orientation distributions of particles in our datasets, and a new Figure S2 to show examples of micrographs and typical 2D classes from each dataset. The FSC curves were already included in Figure S2 (now S3).

Reviewer #4 (Remarks to the Author):

The paper by Bridges et al. is a very careful study of the specific inhibitor, piericidin, binding to complex I. All used different protocols are very well described; their number is impressive. The structure of complex I with bound piericidin was resolved, and it was shown that the inhibitor is bound in the quinone-binding site as expected due to the high similarity of ubiquinone and piericidin molecules.

We thank the reviewer for these positive comments.

Also, the authors claim that the second piericidin-binding site occurs close to the bottom of the quinone-binding channel, and the conclusion of two-site competitive inhibition was made. However, this conclusion probably should be more tentative since the inhibitor-enzyme interaction could be more complicated than described with simple Michaelis-Menten curves, and the number of experimental points for quinone K_m determination should be increased for a precise analysis.

We completely agree that the actual mechanism is very likely more complicated than can be embodied in a simple Michaelis-Menten analysis. However, the model we used (presented in Figure 5b) is not just a simple MM model but an extended form of it – although it produces MM-like curves and allows us to characterise and describe the data using easily appreciated apparent K_M and k_{cat} values (labelled $K_{M,app}$ etc. on Figure 5c). We tested numerous different mechanisms (some of which did not explain our data adequately, Supplementary Table 3) - and have no justification for increasing the model complexity when the simple model in 5b fits our data.

As for the number of experimental points in the K_M determinations, each point for a different Q10 concentration is a separate preparation of proteoliposomes that has to be independently characterised. Increasing the number of data points within a matching dataset is very experimentally challenging, and not as simple as just adding different substrate concentrations to an assay. Furthermore, our modelling analysis were conducted by considering all 32 datapoints simultaneously, to determine the kinetic parameters shown in Figure 5b. Individual values for $K_{M,app}$ (etc.) are presented only to allow for an more intuitive analysis of the apparent trends in parameter values.

The Discussion provides a broad description of the literature data on inhibitor binding however, the analysis of the data obtained seems insufficient. The finding of two piericidin-binding sites arises several questions. The model of two-site competitive binding predicts 10 000 folds difference in the binding tightness. One site is clearly at the top of the ubiquinone binding channel in N2 cluster proximity. If this site is tight-binding, it would be important to compare piericidin and the ubiquinone binding. Since the structures of these compounds are very similar, what particularly makes piericidin to be bound that tight in contrast to ubiquinone?

The piericidin and ubiquinone headgroups differ by the substitution of one of the ketone groups by NH – changing the nature of possible interaction from H-bond acceptor (C=O) to H-bond donor (NH). In this respect piericidin more closely resembles semiquinone than quinone or quinol – although there is no corresponding change in charge or aromaticity, and so we do not refer to piericidin as a SQ analogue in the manuscript. We note that there is no evidence for the release of SQ from the binding site, so it is either very tightly bound or very short lived. Piericidin also differs from ubiquinone by its much shorter tail, which might cause a caging effect (as discussed in Fedor et al. PNAS 2017), and by the hydrophilic hydroxyl group on its tail which is observed near the charged “kink” region of the channel. Furthermore, a second piericidin binding behind the first one (head-to-tail) will help prevent its dissociation. Due to these many differences we are reluctant to extrapolate conclusions about the piericidin binding affinity compared to ubiquinone.

Or could it be that the tight-binding occurs at the second site located partially in ND1? The very interesting and important suggestion that piericidin binding to the second site could be relevant to proton translocation (Discussion, last line) needs some elucidation: how particularly the authors imagine this relevance.

Turning to an explanation for the proposed higher affinity of the second molecule in the phospholipid state, we have now expanded our discussion in the text to clarify how the many subsites of the dispersed second binding are all expected to contribute to the binding affinity: an entropic stabilization of the set of subsites to provide a higher affinity for the site as a whole. We also note that interactions (particularly the potential hydrogen bond) between the two piericidins contribute to the apparent cooperativity. Indeed, the 5 kcal mol⁻¹ interaction energy between piericidins 1 and 2, comparable to a strong hydrogen bond, accounts for a 3000-fold enhanced K_d for piericidin 2 when piericidin 1 is present relative to when it is absent. We appreciate the reviewer’s interest in this second site, and indeed find it very interesting ourselves – but we feel that further speculation on its role in proton pumping is unwarranted, given the very limited amount of experimental evidence available. We have added the following to our text on page 8 accordingly: “However, the distal piericidin does not adopt a single, well-defined binding pose, and its

fluctuations in the MD simulations, together with the unclear density, are better explained by a broad substrate recognition region, rather than a unique, well-defined distal site. The broad recognition region may lead to an entropic stabilization for the set of subsites to provide a higher affinity for the site as a whole, whereas the interaction between the two piericidins may contribute to the cooperativity, explaining the tighter apparent binding of the second piericidin relative to the first.”

The choice of quoting is embarrassing. P.2, 52-57. These lines make the impression that the architecture of the electron transfer chain was not known before publishing the papers cited in the manuscript. However, the position of redox groups (Sazanov’s studies) and the identification of the terminal cluster N2 (Ohnishi’s studies) in the bacterial complex I were known for a long time. Because core subunits of eu- and prokaryotic complex I are highly similar and the structure of the electron transfer chain is almost identical the earliest papers should be cited here. In the discussion of inhibitor binding sites in complex I, the studies of Miyoshi’s group, who made a huge contribution to this issue, are not even mentioned. The paper by Degli-Esposti, who suggested the second piericidin-binding site partially in ND1 (exactly at the same position as in the manuscript), is mentioned but not in this context.

We apologise for omitting to refer to the structure of the enzyme from *Thermus thermophilus* here (although it was referenced later in our text) and have amended our text to read: “The structures illustrate how, as shown previously in the enzyme from *Thermus thermophilus* [Baradaran, 2013], electrons enter...” We agree that Ohnishi’s group led early work to determine the reduction potentials of EPR-visible clusters, but they did not spatially locate the originating clusters, and the reduction potentials are not of central relevance to the work presented here. We apologise for omitting to reference the work of Miyoshi and co-workers as it is obviously pertinent here and have now added references to two Miyoshi papers at the start of the relevant paragraph (page 2). The reviewer is correct that we have not referenced the Degli-Esposti paper for the second binding site of piericidin, but this is because we have referenced the two original papers on this topic by Gutman et al. (from 1970 and 1980).

Minor comments:

p.4, l.140-141. It is not clear whether piericidin binds to the active state of complex I, or it could bind the de-active state but does not prevent conversion the de-active-to-active state. On the other hand, if before piericidin binding the enzyme clearly was in the active state, how the authors know that after inhibitor binding complex I did not convert to the inactive state?

We are confident that we observe piericidin bound to a protein structure essentially identical to the active state. However, we don’t have any data pertaining to whether piericidin could also bind the deactive (or any other) state and subsequently convert it to the active state. For clarity we have amended the sentence on page 4 to read: “Therefore, we observe piericidin bound to the active state of complex I, in which the ubiquinone-binding site is fully configured and ready for catalysis.”

P.6. l.198-218 and Fig. 4. The description of EPR experiment is not clear. The EPR spectrum of the “turnover” sample before and after following thawing and NADH addition should be shown in Fig.4 for comparison with the “inhibited” sample. Were the “turnover” and “inhibited” samples prepared with NADH, Q10 and AOX? Both samples were applied to the Superose column; therefore they should be depleted on NADH, and oxidized by AOX in the presence of oxygen in a way dependent on quinone reductase activity. A comparison of such spectra will show the retention of electron on the N2 cluster due to piericidin binding.

We apologise for the lack of clarity in our presentation of the EPR experiment. The turnover control and piericidin-bound sample were prepared in the same manner as the cryo-EM sample, by the addition of NADH, AOX, DQ, ADH, Catalase and SOD, all of which were removed by separation on the superose 6 column prior to sample freezing. We have added an additional panel with the turnover control data to Figure 4, and amended our text for clarification and to ensure consistent terminology with the figure.

REVIEWERS' COMMENTS

Reviewer #1 (Remarks to the Author):

The authors have carefully addressed the comments of all reviewers. I have no further comments.

Reviewer #2 (Remarks to the Author):

I thank the authors for addressing all my comments
I believe the current version is ready for publication

Reviewer #4 (Remarks to the Author):

The manuscript was amended significantly. A number of requested explanations were introduced. Now it is clear that this study presents a remarkable example that a resolved structure does not always reflect the true enzyme state. The kinetic measurements clearly indicate that from two piericidin-binding sites, the second site, distant from N2, is bound much more tightly than the first one. This finding suggests high stability of the second site surrounding, but no piericidin was present in this site in the resolved structure. In contrast, only the first site was found occupied with piericidin. Following the model, piericidin is weakly bound in the first site and therefore should be washed out upon delipidation as ubiquinone, which never was shown in resolved so far structures of complex I. The authors explain such discrepancy by the difference between membrane-bound and solubilized enzyme (the last phrase in Results section). Then the question arises: in which extent the solubilized enzyme maintains all properties of the membrane-bound? How the distal part of quinone-binding channel could be distorted upon solubilization? From my point of view these questions are important and should be considered in Discussion.

The minor comments.

The statement that piericidin can be trapped in local energy minima as suggested for short chain ubiquinones Q1 and Q2 (last paragraph in Discussion) does not look solid because the alkyl chain of piericidin is significantly longer.

The structure of the membrane-bound enzyme is impossible to resolve so far, but it is possible to perform kinetic studies using the solubilized complex I and find out whether, in this case, the results will correlate with the resolved structure. Maybe, this is a good point for future studies.

Reviewer #1 (Remarks to the Author):

The authors have carefully addressed the comments of all reviewers. I have no further comments.

Reviewer #2 (Remarks to the Author):

I thank the authors for addressing all my comments. I believe the current version is ready for publication

Reviewer #4 (Remarks to the Author):

The manuscript was amended significantly. A number of requested explanations were introduced. Now it is clear that this study presents a remarkable example that a resolved structure does not always reflect the true enzyme state. The kinetic measurements clearly indicate that from two piericidin-binding sites, the second site, distant from N2, is bound much more tightly than the first one. This finding suggests high stability of the second site surrounding, but no piericidin was present in this site in the resolved structure. In contrast, only the first site was found occupied with piericidin. Following the model, piericidin is weakly bound in the first site and therefore should be washed out upon delipidation as ubiquinone, which never was shown in resolved so far structures of complex I. The authors explain such discrepancy by the difference between membrane-bound and solubilized enzyme (the last phrase in Results section). Then the question arises: in which extent the solubilized enzyme maintains all properties of the membrane-bound? How the distal part of quinone-binding channel could be distorted upon solubilization? From my point of view these questions are important and should be considered in Discussion.

We thank the reviewer for these insightful comments. Indeed, we have spent a lot of “coffee time” speculating on the implications of the discrepancy between the membrane-bound and detergent-solubilized enzymes that this result implies. Of course, the detergent-solubilized enzyme can never be regarded as identical to the enzyme in the mitochondrion, and we have to work with that challenge. We believe we have captured this point sufficiently well in the closing paragraph of our results, where we state:

“Our data are consistent with two piericidins binding in our membrane-bound enzyme assay, and with lower occupancy of the distal site in the detergent-solubilised, structurally-characterised enzyme. Specific phospholipids are known to be essential for complex I activity³⁵ and their loss and/or the presence of detergent may destabilize the second piericidin and affect the properties of the distal section of the ubiquinone-binding channel. Although our results thus present an example of how the structurally-characterized detergent-solubilized protein differs from the membrane-bound enzyme, our interpretation is consistent with an important role for phospholipids, and cardiolipin in particular, in complex I catalysis^{35,36}.”

The minor comments.

The statement that piericidin can be trapped in local energy minima as suggested for short chain ubiquinones Q1 and Q2 (last paragraph in Discussion) does not look solid because the alkyl chain of piericidin is significantly longer.

We have now clarified that piericidin has an alkyl chain length similar to Q3 - we believe this is sufficiently short that it may be subject to similar ‘trapping’ behaviour as Q1 and Q2.

The structure of the membrane-bound enzyme is impossible to resolve so far, but it is possible to perform kinetic studies using the solubilized complex I and find out whether, in this case, the results will correlate with the resolved structure. Maybe, this is a good point for future studies.

We thank the reviewer for this interesting future suggestion. Indeed, we have considered undertaking inhibitor kinetics studies with the isolated enzyme, but have been deterred so far because such studies suffer from the additional challenge of requiring hydrophilic quinone analogues which (along with the inhibitor) exhibit complex behaviour due to partitioning between different phases (aqueous and detergent/phospholipid/ubiquinone micelles) and which may also enter the ubiquinone-binding channel in series. In the meantime, we note the earlier studies on stoichiometry of piericidin binding in both situations that provide a basis for direct comparison in support of our proposals.